# A positive charge region of *Salmonella* FliI is required for ATPase formation and efficient flagellar protein export

Miki Kinoshita [1], Keiichi Namba [1,2,3] & Tohru Minamino [1✉]

The FliH$_2$FliI complex is thought to pilot flagellar subunit proteins from the cytoplasm to the transmembrane export gate complex for flagellar assembly in *Salmonella enterica*. FliI also forms a homo-hexamer to hydrolyze ATP, thereby activating the export gate complex to become an active protein transporter. However, it remains unknown how this activation occurs. Here we report the role of a positively charged cluster formed by Arg-26, Arg-27, Arg-33, Arg-76 and Arg-93 of FliI in flagellar protein export. We show that Arg-33 and Arg-76 are involved in FliI ring formation and that the *fliI(R26A/R27A/R33A/R76A/R93A)* mutant requires the presence of FliH to fully exert its export function. We observed that gain-of-function mutations in FlhB increased the probability of substrate entry into the export gate complex, thereby restoring the export function of the Δ*fliH fliI(R26A/R27A/R33A/R76A/R93A)* mutant. We suggest that the positive charge cluster of FliI is responsible not only for well-regulated hexamer assembly but also for substrate entry into the gate complex.

[1] Graduate School of Frontier Biosciences, Osaka University, Suita, Osaka, Japan. [2] RIKEN SPring-8 Center and Center for Biosystems Dynamics Research, Suita, Osaka, Japan. [3] JEOL YOKOGUSHI Research Alliance Laboratories, Osaka University, Suita, Osaka, Japan. ✉email: tohru@fbs.osaka-u.ac.jp

The flagellum of *Salmonella enterica* (hereafter referred to as *Salmonella*) is a supramolecular motility machine consisting of the basal body, the hook, and the filament[1]. For construction of the flagellum on the cell surface, a specialized protein export apparatus uses ATP and a proton motive force (PMF) across the cytoplasmic membrane to transport flagellar building blocks from the cytoplasm to the distal end of the growing flagellar structure[2]. The flagellar export apparatus is composed of a PMF-driven transmembrane export gate complex made of FlhA, FlhB, FliP, FliQ, and FliR and a cytoplasmic ATPase ring complex consisting of FliH, FliI, and FliJ (Fig. 1)[3,4].

The PMF-driven export gate complex is located inside the basal body MS ring formed by a transmembrane protein, FliF (Fig. 1)[5]. FliP, FliQ, and FliR form a tubular structure with a stoichiometry of five FliP, four FliQ, and one FliR with a helical symmetry similar to those of the rod, hook, and filament, and the central pore of this $FliP_5FliQ_4FliR_1$ complex is thought to be a polypeptide channel for the translocation of flagellar building blocks across the cytoplasmic membrane[6–8]. The N-terminal transmembrane domain of FlhB ($FlhB_{TM}$) associates with the $FliP_5$-$FliQ_4FliR_1$ complex to form the $FliP_5FliQ_4FliR_1FlhB_1$ complex and is proposed to coordinate opening of the export gate for substrate entry into the polypeptide channel[9]. FlhA assembles into a homo-nonamer through its C-terminal cytoplasmic domain ($FlhA_C$)[10,11] and forms a pathway for the transit of

protons across the cytoplasmic membrane[12,13]. $FlhA_C$ and the C-terminal cytoplasmic domain of FlhB ($FlhB_C$) project into the cytoplasmic cavity of the basal body C ring and form a docking platform for the cytoplasmic ATPase complex, flagellar export chaperones, and flagellar building blocks[14].

The cytoplasmic ATPase ring complex is composed of 12 copies of FliH, 6 copies of FliI, and a single copy of FliJ (Fig. 1)[15,16]. FliI forms a homo-hexamer to hydrolyze ATP at an interface between FliI subunits[17–19]. The FliI ring structure is localized to the flagellar base through interactions of the extreme N-terminal region of FliH ($FliH_{EN}$) with FlhA and a C ring protein, FliN[20–22]. FliJ binds to the center of the FliI ring[15] and plays an important role in an ATP-dependent gate activation mechanism[23,24].

FliI also exists free in the cytoplasm as a heterotrimeric complex formed with a FliH dimer (Fig. 1)[25]. This $FliH_2FliI$ complex binds to FliJ, export substrates and flagellar chaperone/substrate complexes, and facilitates their docking to the $FlhA_C$-$FlhB_C$ docking platform for efficient and robust flagellar protein export[26–29]. This has been verified by in vitro reconstitution experiments using inverted membrane vesicles[30]. High-resolution live cell imaging using FliI labeled with a yellow fluorescent protein (FliI-YFP) has shown that not only the FliI ring structure but also several $FliH_2FliI$ complexes bind to the flagellar base through interactions of FliH with FliN and FlhA (Fig. 1)[31]. Because FliI-YFP shows turnover between the basal body and the cytoplasmic pool, the $FliH_2FliI$ complex is thought to act as a dynamic carrier to escort FliJ and export substrates and chaperone/substrate complexes from the cytoplasm to the $FlhA_C$-$FlhB_C$ docking platform[31].

The N-terminal domain of FliI ($FliI_N$) is responsible not only for the interaction with FliH but also for $FliI_6$ ring formation[32]. The C-terminal domain of FliH ($FliH_C$) binds to the extreme N-terminal α-helix consisting of residues 2–21 of FliI ($FliI_{EN}$) and a positively charged cluster formed by Arg-26, Arg-27, Arg-30, Arg-33, Arg-76, and Arg-93 of $FliI_N$ (Fig. 2a)[16]. Because the binding of $FliH_C$ to $FliI_{EN}$ not only inhibits $FliI_6$ ring formation but also reduces the FliI ATPase activity considerably[25,33], FliH seems to play a regulatory role in the energy coupling mechanism of the flagellar protein export apparatus. However, it remains unknown how FliH regulates ATP hydrolysis by the FliI ATPase for flagellar protein export. To clarify this question, we performed mutational analysis of the positively charged cluster of $FliI_N$. We show that Arg-33 and Arg-76 of $FliI_N$ are involved in well-regulated FliI ring formation. We also show that this positive charge cluster is required for efficient entry of flagellar building blocks into the transmembrane export gate complex.

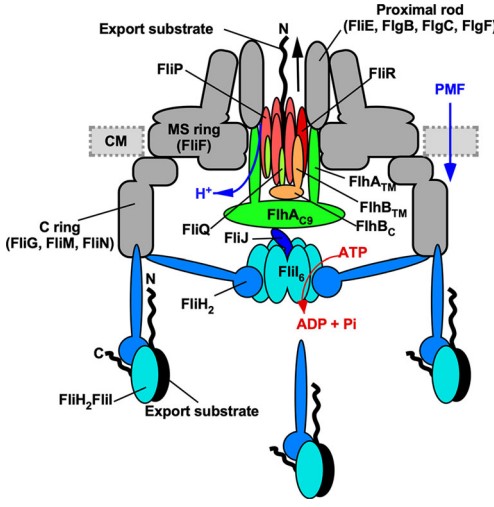

**Fig. 1 Schematic diagram of the flagellar protein export apparatus.** The flagellar protein export apparatus is composed of a transmembrane export gate complex made of FlhA, FlhB, FliP, FliQ, and FliR and a cytoplasmic ATPase ring complex consisting of FliH, FliI, and FliJ. The export gate complex is located inside the MS ring and utilizes proton motive force (PMF) across the cytoplasmic membrane (CM) to drive proton ($H^+$)-coupled flagellar protein export. FliP, FliQ, and FliR form a polypeptide channel complex. The N-terminal transmembrane domain of FlhB ($FlhB_{TM}$) associates with the FliP/FliQ/FliR complex, and its C-terminal cytoplasmic domain ($FlhB_C$) projects into the central cavity of the C ring. FlhA forms a homo-nonamer through interactions between the C-terminal cytoplasmic domain of FlhA ($FlhA_C$), and its N-terminal transmembrane domain ($FlhA_{TM}$) acts as a transmembrane $H^+$ channel. The cytoplasmic ATPase ring complex associates with the C ring through an interaction between FliH and a C ring protein, FliN. ATP hydrolysis by the FliI ATPase activates the export gate complex through an interaction between FliJ and $FlhA_C$, allowing the gate complex to become an active protein transporter to couple the proton flow through the FlhA proton channel to the translocation of export substrates into the polypeptide channel of the FliP/FliQ/FliR complex. The $FliH_2FliI$ complex is thought to act as a dynamic carrier to bring export substrates from the cytoplasm to the export gate complex.

## Results

**Effect of alanine substitution in the positive charge cluster on motility.** Arg-26, Arg-27, Arg-30, Arg-33, Arg-76, and Arg-93 of *Salmonella* FliI ATPase form a positive charge cluster on the molecular surface of the FliI hexamer[18], and Arg-26, Arg-30, and Arg-33 are relatively well conserved among FliI homologs (Fig. 2b). To clarify the role of these Arg residues in flagellar protein export, we constructed the following nine *fliI* mutants: five mutants with a single mutation, *fliI(R26A)*, *fliI(R27A)*, *fliI(R30A)*, *fliI(R33A)*, and *fliI(R93A)*, and four mutants with multiple mutations, *fliI(R26A/R27A)*, *fliI(R26A/R27A/R33A)*, *fliI(R26A/R27A/R33A/R76A)*, and *fliI(R26A/R27A/R33A/R76A/R93A)* (hereafter referred to as *fliI-2A*, *fliI-3A*, *fliI-4A*, and *fliI-5A*, respectively). All of these mutant variants except FliI-5A fully restored the motility of a *Salmonella* Δ*fliI* mutant when they were expressed even at a relatively low copy level from pET19b-based plasmids (Supplementary Fig. 1a). In contrast, the motility of the *fliI-5A* mutant was lower than that of

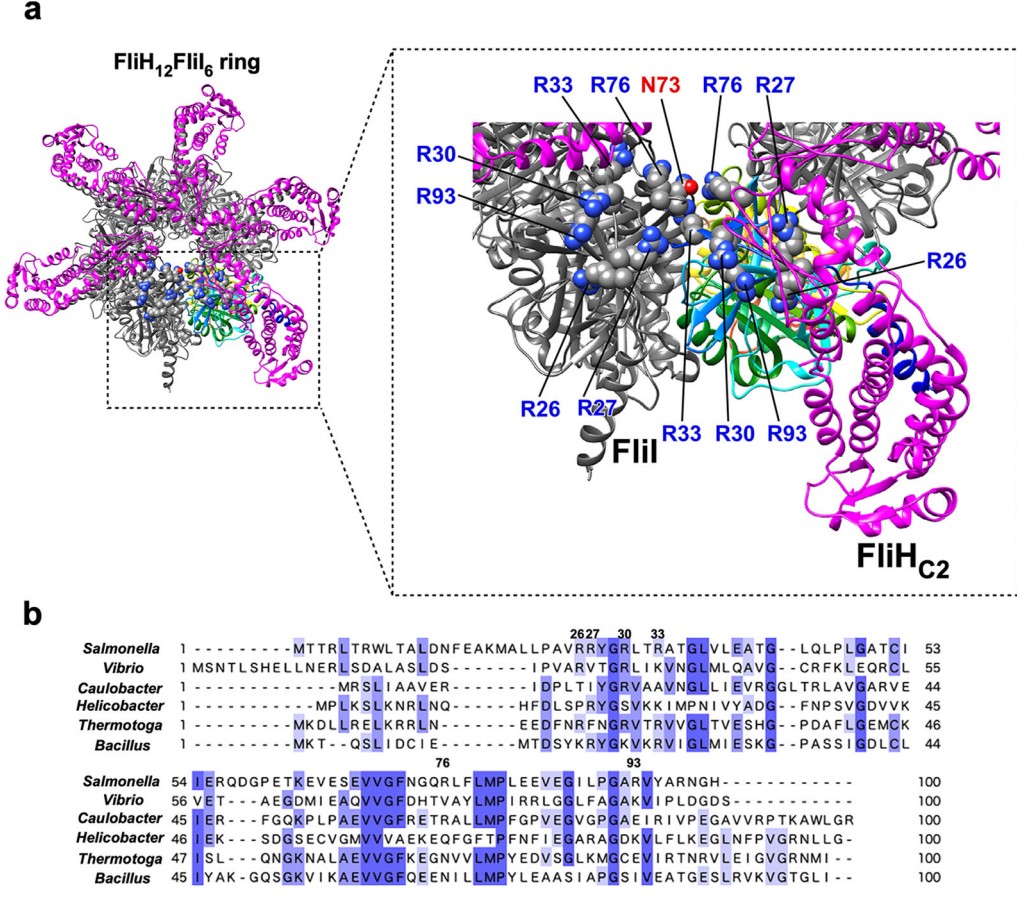

**Fig. 2 Structural model for the cytoplasmic ATPase ring complex. a** Cα ribbon representation of the FliH₂FliI complex (PDB ID: 5B0O). Arg-26 (R26), Arg-27 (R27), Arg-30 (R30), Arg-33 (R33), Arg-76 (R76), and Arg-93 (R93) form a positive charge cluster on the molecular surface of the N-terminal domain of FliI. The C-terminal domain of FliH forms a dimer (FliH$_{C2}$) to bind to the N-terminal α-helix consisting of residues 2–21 of FliI and the positive charge cluster. Arg-33 and Arg-76 are located at an interface between FliI subunits, and Arg-76 is in very close proximity of Asn-73 (N73) of its nearest FliI subunit. **b** Multiple sequence alignment of the N-terminal domain of FliI orthologs. Multiple sequence alignment was carried out by Clustal Omega. Conserved residues are labeled with blue colors. UniProt Accession numbers: *Salmonella*, P26465; *Vibrio*, A0A1Q1PM95; *Caulobacter*, B8H363; *Helicobacter*, O07025, *Thermotoga*, A0A2N5RHQ8; *Bacillus*, P23445.

wild-type cells (Fig. 3a). Consistently, the *fliI-5A* mutation reduced the secretion level of FlgD (Fig. 3b). Because the *fliI-5A* mutation did not affect the steady cellular level of FliI (Supplementary Fig. 1b), we conclude that this mutation affects the protein export function of FliI.

To investigate whether the overexpression of FliI-5A improves the motility, we cloned this *fliI* allele into the pTrc99A vector. When FliI-5A was overexpressed from a pTrc99A-based plasmid, FliI-5A restored the motility of the Δ*fliI* mutant to the wild-type level (Fig. 3a). Consistently, the secretion level of FlgD was also at the wild-type level (Fig. 3b), indicating that the reduced export activity of FliI-5A is restored to the wild-type level by an increase in its expression level.

**Effect of alanine substitution in the positive charge cluster on FliH–FliI interaction.** The positive charge cluster of FliI$_N$ is involved in the interaction with FliH$_C$ (Fig. 2a)[16]. To test whether the *fliI-5A* mutation reduces the binding affinity of FliI for FliH, we carried out pull-down assays by Ni affinity chromatography (Supplementary Fig. 2). Untagged FliH co-purified with His-FliI, in agreement with a previous report[25]. In contrast, the *fliI-5A* mutation reduced the binding affinity of FliI for FliH (Supplementary Fig. 2). Although the *fliI-2A*, *fliI-3A* and *fliI-4A* mutations did not affect motility at all (Supplementary Fig. 1a), these three mutations also reduced the binding affinity of FliI for FliH

(Supplementary Fig. 2). Therefore, we conclude that electrostatic interactions between the positive charge cluster of FliI$_N$ and FliH$_C$ are dispensable for flagellar protein export although these interactions stabilize an interaction between FliI$_{EN}$ and FliH$_C$. Because a complete loss of the positive charges by *fliI-5A* mutation reduced the protein transport activity (Fig. 3), we hypothesize that this positive charge cluster may be involved in the interaction with other export apparatus components for efficient flagellar protein export.

**Multicopy effect of FliI(R33A) and FliI-3A on *Salmonella* cell growth.** When FliI(R33A) and FliI-3A (R26A/R27A/R33A triple mutation) were expressed at a relatively high copy level from pTrc99A-based plasmids, the motility ring of these two *fliI* mutants were smaller than the wild-type ring (Supplementary Fig. 1c). However, they fully restored the motility of the Δ*fliI* mutant when they were expressed at a relatively low copy level from pET19b-based plasmids (Supplementary Fig. 1a). As the motility ring size also depends on the growth rate of *Salmonella* motile cells, we investigated whether the higher expression level of FliI with either R33A or FliI-3A substitution reduces the growth rate of *Salmonella* cells. When FliI(R33A) and FliI-3A were expressed from pET19b-based plasmids, these two FliI mutants did not affect the cell growth at all (Supplementary Fig. 3a). In contrast, when FliI(R33A) and FliI-3A were expressed

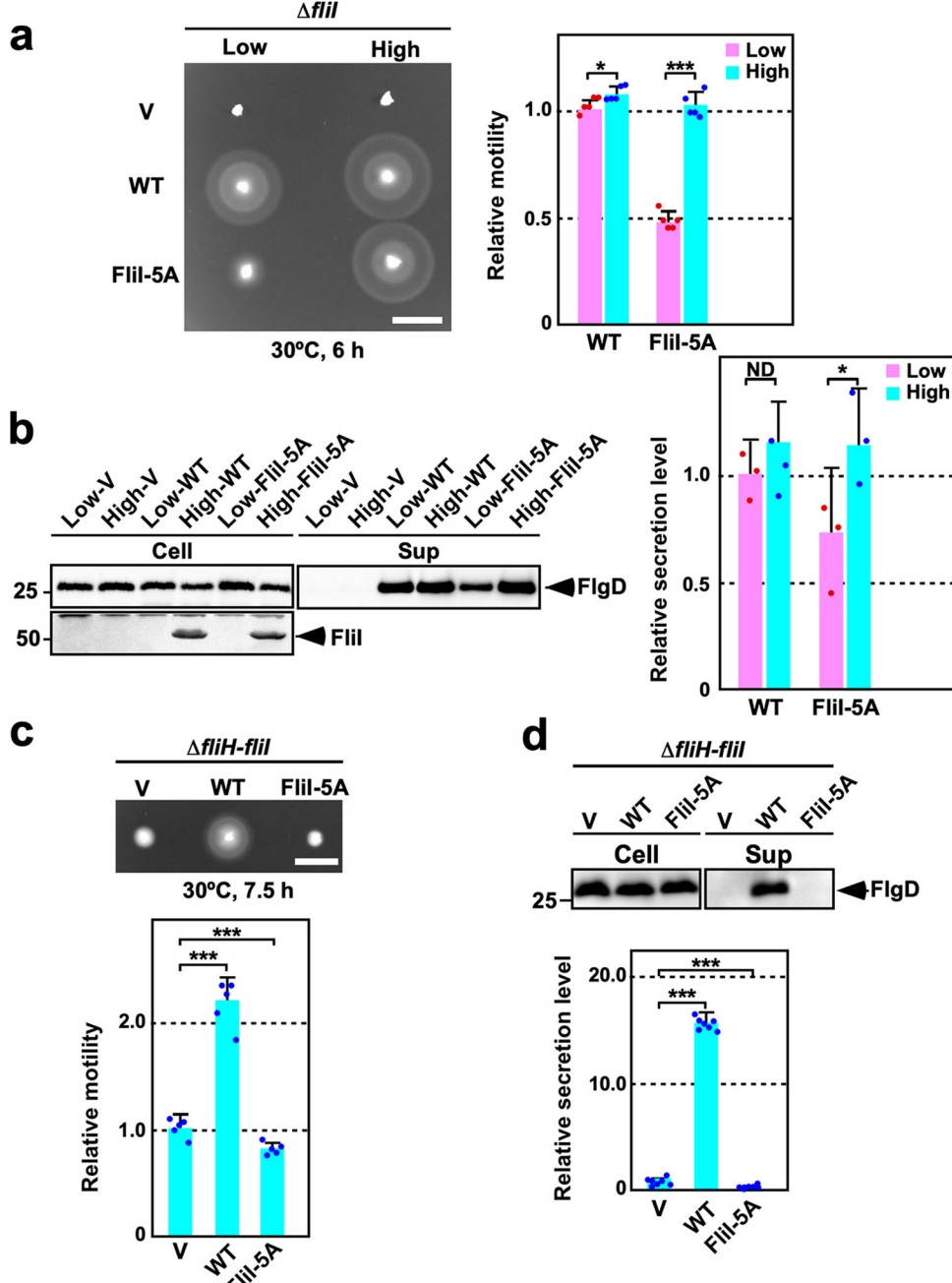

from pTrc99A-based plasmids, they reduced the cell growth (Supplementary Fig. 3a) even though their expression levels were lower than the wild-type level (Supplementary Fig. 3b). To confirm this, we measured the growth rate of *Salmonella* cells overexpressing wild-type FliI, FliI(R33A), or FliI-3A (Fig. 4a). Overexpression of wild-type FliI slightly reduced the cell growth rate compared to the vector control. In contrast, cell growth was totally inhibited by induction of FliI(R33A) and FliI-3A (Fig. 4a), even though their expression level was lower than the wild-type level (Fig. 4b).

FliH suppresses FliI ring formation in solution, thereby suppressing the ATPase activity of FliI[25,33]. Therefore, we next tested whether FliH relieves the growth inhibition caused by the *fliI(R33A)* and *fliI-3A* mutations. When FliI(R33A) or FliI-3A was co-expressed with FliH from the pTrc99A-based plasmids, no growth inhibition occurred (Supplementary Fig. 3a) although their expression levels were increased (Supplementary Fig. 3b, c). As

described above, the *fliI-3A* mutation reduced the binding affinity of FliI for FliH, whereas the R33A mutation did not (Supplementary Fig. 2). Because FliH strongly binds to FliI$_{EN}$ to suppress FliI ring formation[16,25,32], we suggest that the binding of FliH to FliI$_{EN}$ inhibits premature FliI ring formation in the cytoplasm, thereby not only suppressing the growth inhibition caused by these two *fliI* mutations but also increasing the steady cellular level of FliI (R33A) and FliI-3A (Supplementary Fig. 3b, c).

**Multicopy effect of the *fliI-3A* and *fliI-4A* mutations on the ATPase activity of FliI.** The ATPase activity of FliI displays a protein concentration-dependent positive cooperativity, indicating that FliI ring formation is required for efficient ATP hydrolysis by the FliI ATPase[17,33]. This raises the possibility that the *fliI (R33A)* and *fliI-3A* mutations increase the probability of premature FliI ring formation in the cytoplasm, thereby reducing the cytoplasmic ATP level enough to inhibit the cell growth. To test

**Fig. 3 Characterization of the *fliI-5A* mutant. a** Motility of MKM30 (Δ*fliI*) cells transformed with pET19b (indicated as Low copy, V), pTrc99A (indicated as High copy, V), pMM1701 (pET19b/His-FliI, indicated as Low copy, WT), pMM1702 (pTrc99A/His-FliI, indicated as High copy, WT), pMM1701-5A [indicated as pET19b/His-FliI(R26A/R27A/R33A/R76A/R93A), Low copy, FliI-5A], or pMM1702-5A [pTrc99A/His-FliI(R26A/R27A/R33A/R76A/R93A), indicated as High copy, FliI-5A] in 0.35% soft agar plates containing 100 µg ml⁻¹ ampicillin. The plate was incubated at 30 °C for 6 h. The diameter of the motility ring of five colonies of each transformant was measured. The average diameter of the motility ring of MKM30 cells harboring pMM1701 was set to 1.0, and then relative diameter of the motility ring was calculated. Dots indicate individual data points. Vertical bars indicate standard deviations. Scale bar, 1.0 cm. **b** Effect of the *fliI-5A* mutation on flagellar protein export. The above transformants were exponentially grown in L-broth, and then whole cellular (Cell) and culture supernatant (Sup) fractions were prepared, followed by SDS-PAGE and finally immunoblotting with polyclonal anti-FlgD (first row) or anti-FliI (second row) antibody. The positions of molecular mass markers are indicated on the left. The regions of interest were cropped from original immunoblots shown in Supplementary Fig. 9a. The density of each FlgD band on immunoblots is normalized for the cellular FlgD levels. These data are the average of four independent experiments. Dots indicate individual data points. Vertical bars indicate standard deviations. **c** Effect of the *fliI-5A* mutation on motility in the absence of FliH. The diameter of the motility ring of 5 colonies of MMHI001 (Δ*fliH-fliI*) cells carrying with pTrc99A (V), pMM1702 (WT), or pMM1702-5A (FliI-5A) was measured. The average diameter of the motility ring of the vector control was set to 1.0, and then relative diameter of the motility ring of each transformant was calculated. Dots indicate individual data points. Vertical bars indicate standard deviations. Scale bar, 0.5 cm. **d** Effect of the *fliI-5A* mutation on flagellar protein export in the absence of FliH. The above transformants were exponentially grown in L-broth, and then whole cellular and culture supernatant fractions were analyzed by immunoblotting with polyclonal anti-FlgD antibody. The regions of interest were cropped from original immunoblots shown in Supplementary Fig. 9b. Relative secretion levels of FlgD were measured. These data are average of seven independent experiments. Dots indicate individual data points. Vertical bars indicate standard deviations. Comparisons between datasets were performed using a two-tailed Student's *t* test. A *P* value of <0.05 was considered to be statistically significant difference. *$P < 0.05$; ***$P < 0.001$; ND, no statistical difference.

this, we measured the ATPase activity of FliI(R33A) and FliI-3A proteins at several distinct protein concentrations (Fig. 4c and Table 1). The ATPase activity of FliI increased from $0.001 \pm 0.0001$ to $0.888 \pm 0.222$ [mean ± standard deviation (SD), $n = 3$] nmol of phosphate min⁻¹ µg⁻¹ with an increase in the protein concentration from 17 to 680 nM. The ATPase activity of FliI(R33A) and FliI-3A also showed a protein concentration-dependent positive cooperativity (Fig. 4c and Table 1) but was greatly higher than that of the wild type even at a very low protein concentration. At FliI concentration of 85 nM, the ATPase activity of wild-type FliI, FliI(R33A), and FliI-3A was $0.008 \pm 0.001$, $1.211 \pm 0.09$, and $1.898 \pm 0.036$ nmol of phosphate min⁻¹ µg⁻¹, respectively.

To test whether FliI-3A forms homo-hexamer, we analyzed the ability of FliI ring formation in the presence of $Mg^{2+}$-ADP-AlF₄, which is a non-hydrolyzable ATP analog (Fig. 5). At a FliI concentration of 1 µM, many ring-like particles were observed in the FliI-3A sample but not in the wild-type FliI one. Only end-on images were selected, aligned in a reference-free mode and averaged, which yielded a ring structure with an approximate sixfold symmetry (Fig. 5, inset). Because the *fliI(R26A)*, *fliI(R27A)*, and *fliI-2A* (R26A/R27A double mutation) mutations did not affect the cell growth, we suggest that the *fliI(R33A)* mutation increases the probability of autonomous FliI ring formation even at a low protein concentration.

FliI-4A also contains the R26A/R27A/R33A triple mutation in addition to the R76A substitution, but overexpression of FliI-4A did not inhibit the cell growth in a way similar to wild-type FliI (Fig. 4a), raising the possibility that the *fliI(R76A)* substitution may reduce the ring formation efficiency of FliI-3A. To clarify this possibility, we purified FliI-4A and measured its ATPase activity at different protein concentrations (Fig. 4c and Table 1). The ATPase activity of FliI-4A was $0.021 \pm 0.004$ nmol of phosphate min⁻¹ µg⁻¹ at a protein concentration of 85 nM. This ATPase activity of FliI-4A was higher than the wild-type level at the same protein concentration but was much lower than that of FliI-3A. Consistently, FliI-4A formed a hexamer but the ring formation efficiency was lower than FliI-3A albeit higher than the wild type (Fig. 5). These suggest that the *fliI(R76A)* substitution partially suppresses efficient FliI ring formation caused by the *fliI(R33A)* mutation, thereby suppressing the growth inhibition caused by overexpression of FliI-3A. Because Arg-33 and Arg-76 are located at an interface between FliI subunits, and Arg-76

forms a hydrogen bond with Asn-73 of its neighboring subunit (Fig. 2a), we suggest that Arg-33, Asn-73, and Arg-76 are critical for well-regulated FliI ring formation.

**FliH deletion effect on motility of *fliI* mutants.** To clarify the role of the positive charge cluster of FliI in flagellar protein export, we investigated motility of *fliI* mutants in the absence of FliH. Because overexpression of FliI bypasses the FliH defect to a significant degree[34,35], we transformed a *Salmonella fliH-fliI* double null mutant (Δ*fliH-fliI*) with pTrc99A-based plasmids encoding the FliI mutant variants we studied above and analyzed the motility of the resulting transformants. Except for FliI(R30A) and FliI(R33A), single alanine substitutions in the positive charge cluster of FliI_N reduced motility in soft agar (Supplementary Fig. 4a, left panel). Consistently, the secretion level of FlgD was lower than the wild-type level (Supplementary Fig. 4b). Interestingly, the *fliI-2A* mutation reduced flagella-driven motility compared to the R26A and R27A single mutations, and the R33A mutation improved the motility defect of the *fliI-2A* mutant to a considerable degree (Supplementary Fig. 4a, right panel). A complete loss of the positive charges of FliI by *fliI-5A* mutation inhibited motility of and the secretion of FlgD by the Δ*fliH-fliI* mutant (Fig. 3c, d), suggesting that FliI-5A exerts an inhibitory effect on flagellar protein export in the absence of FliH. These observations suggest that the positive charge cluster of FliI_N is involved in well-regulated flagellar protein export.

The FliI ATPase plays an important role in substrate recognition[26–28]. Addition of purified FliH₂FliI complex at a final concentration of 1.5 µM to the in vitro assay solution increases the level of FlgD transported to the inside of inverted membrane vesicles by 20-fold, indicating that the FliH₂FliI complex facilitates the export of FlgD[30]. To clarify why the *fliI-5A* mutation abolishes the secretion of FlgD in the absence of FliH, we analyzed the FliI–FlgD interaction by glutathione S-transferase (GST) affinity chromatography. A very small amount of FlgD was detected in the elution fractions derived from GST alone presumably due to its non-specific binding to the column (Fig. 6a). In contrast to GST alone, much higher amounts of FlgD co-purified with GST-FliI (Fig. 6b), indicating a specific interaction between FliI and FlgD. The *fliI-5A* mutation did not inhibit the interaction of FliI with FlgD (Fig. 6c).

FliI also interacts with FlhA_C, FlhB_C, and FliJ during flagellar protein export[14]. To investigate whether the *fliI-5A* mutation

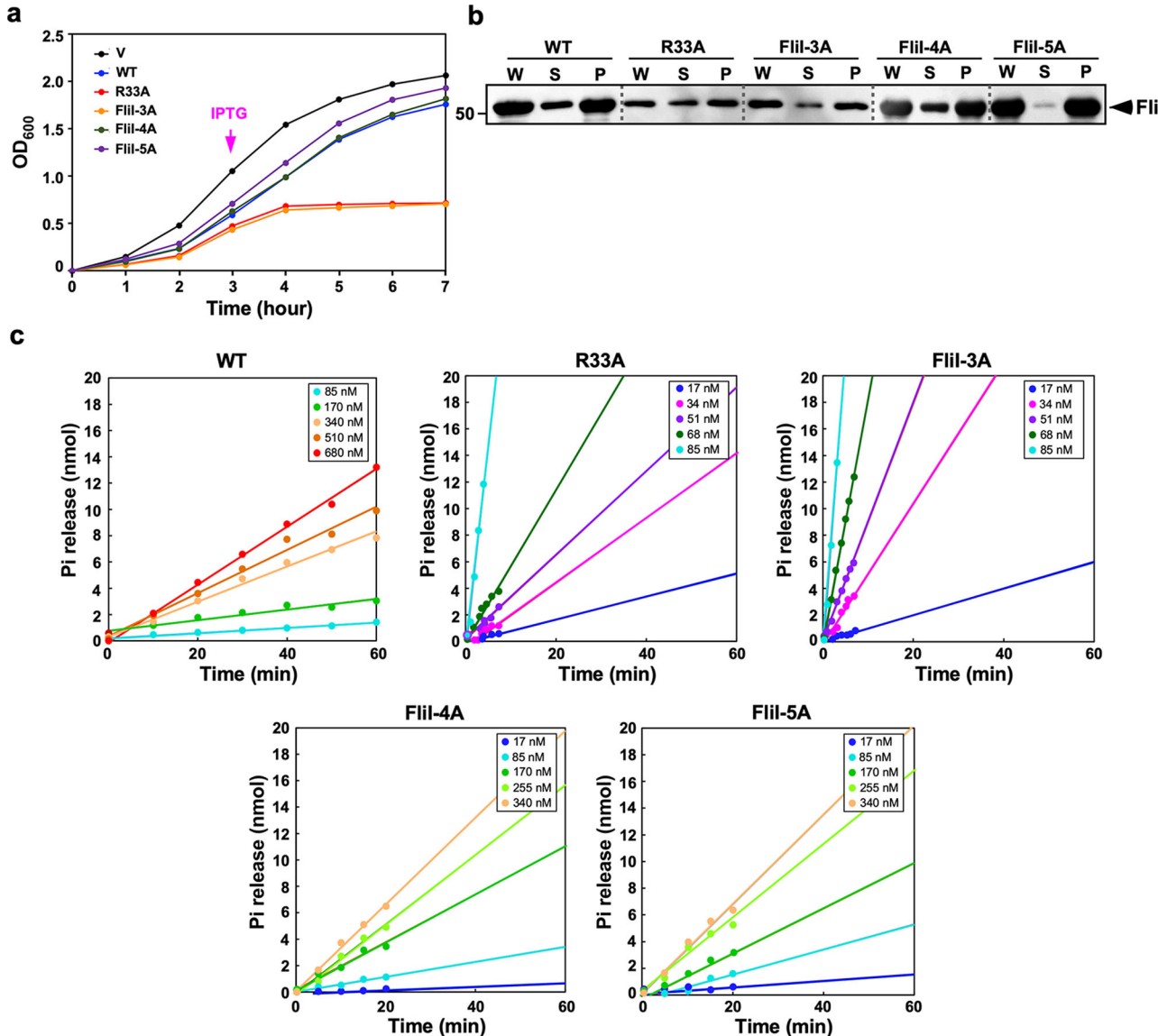

**Fig. 4 Multicopy effect of FliI mutant proteins on cell growth. a** Growth curve of MKM30 (Δ*fliI*) cells transformed with pTrc99A (V), pMM1702 (WT), pMM1702(R33A) [pTrc99A/His-FliI(R33A), indicated as R33A], pMM1702-3A [pTrc99A/His-FliI(R26A/R27A/R33A), indicated as FliI-3A], pMM1702-4A [pTrc99A/His-FliI(R26A/R27A/R33A/R76A), indicated as FliI-4A], or pMM1702-5A [pTrc99A/His-FliI(R26A/R27A/R33A/R76A/R93A), indicated as FliI-5A]. Cells were grown in L-broth containing ampicillin at 30 °C for 3 h and then IPTG was added at a final concentration of 0.1 mM. The $OD_{600}$ value of each culture was monitored every hour. These data are the average of three independent biological replicates. The experimental errors are within 10%. **b** Effect of the *fliI(R3A)*, *fliI-3A*, *fliI-4A*, and *fliI-5A* mutations on the expression level of FliI. Immunoblotting, using polyclonal FliI antibody, of whole cellular (W), soluble (S), and insoluble (P) fractions prepared from the above transformants. The regions of interest were cropped from original immunoblots shown in Supplementary Fig. 10. **c** Effect of FliI mutations on dependence of the FliI ATPase activity on protein concentration. The ATPase activity of purified His-FliI (WT), His-FliI(R33A) (R33A), His-FliI(R26A/R27A/R33A) (FliI-3A), His-FliI(R26A/R27A/R33A/R76A) (FliI-4A), or His-FliI(R26A/R27A/R33A/R76A/R93A) (FliI-5A) was measured at different protein concentrations in the presence of 4 mM ATP by using the Malachite Green assay. The activity is expressed as nmol of Pi released per min per μg of FliI.

affects the interactions of FliI with these three proteins, we performed pull-down assays by GST affinity chromatography. Small amounts of FliI co-purified with GST-FlhA$_C$, GST-FlhB$_C$, and GST-FliJ but not with GST alone (Fig. 7a). Furthermore, compared to the GST control, elution of FliI was clearly observed as a delayed wash out, reflecting weak and highly dynamic interactions of FliI with GST-FlhA$_C$, GST-FlhB$_C$, and GST-FliJ (Fig. 7a). Two-tailed Student's *t* tests revealed that the amounts of FliI-5A co-purified with GST-FlhB$_C$ and GST-FliJ were significantly higher than those of wild-type FliI ($p < 0.05$; Fig. 7b), indicating that the *fliI-5A* mutation increases the

binding affinities of FliI for FlhB$_C$ and FliJ. In contrast, this *fliI-5A* mutation reduced the binding affinity of FliI for FlhA$_C$ ($p < 0.05$; Fig. 7b). Because the level of FlgD secreted by the Δ*fliH fliI-5A* mutant was lower than that by the Δ*fliH-fliI* mutant (Fig. 3d), we suggest that FliI-5A may bind to FlhB$_C$ and FliJ to block the flagellar protein export process in the absence of FliH. Since the overexpression of FliI-5A restored motility to the wild-type level in the presence of FliH (Fig. 3a), we propose that an interaction between the positive charge cluster of FliI and FlhA$_C$ may be involved in the flagellar protein export process.

**Table 1 ATPase activity of FliI mutants.**

| Protein concentration (nM) | FliI ATPase activity (nmol of phosphate min$^{-1}$ μg$^{-1}$) (mean ± SD) | | | | |
|---|---|---|---|---|---|
| | Wild type | R33A | FliI-3A | FliI-4A | FliI-5A |
| 17 | 0.001 ± 0.0001 | 0.008 ± 0.002 | 0.011 ± 0.004 | 0.002 ± 0.001 | 0.001 ± 0.002 |
| 34 | 0.0014 ± 0.0001 | 0.041 ± 0.001 | 0.075 ± 0.019 | 0.0033 ± 0.002 | 0.0065 ± 0.003 |
| 51 | N.D. | 0.079 ± 0.001 | 0.231 ± 0.007 | N.D. | N.D. |
| 68 | 0.0019 ± 0.001 | 0.155 ± 0.049 | 0.643 ± 0.017 | 0.0086 ± 0.001 | 0.0083 ± 0.002 |
| 85 | 0.008 ± 0.001 | 1.211 ± 0.090 | 1.898 ± 0.036 | 0.021 ± 0.004 | 0.024 ± 0.012 |
| 170 | 0.045 ± 0.015 | N.D. | N.D. | 0.115 ± 0.050 | 0.093 ± 0.033 |
| 255 | N.D. | N.D. | N.D. | 0.254 ± 0.104 | 0.196 ± 0.052 |
| 340 | 0.258 ± 0.049 | N.D. | N.D. | 0.467 ± 0.114 | 0.333 ± 0.049 |
| 510 | 0.527 ± 0.163 | N.D. | N.D. | N.D. | N.D. |
| 680 | 0.888 ± 0.222 | N.D. | N.D. | N.D. | N.D. |

The ATPase activity of each FliI protein was measured using the Malachite Green assay. These data are the average of three independent measurements.
N.D. not determined.

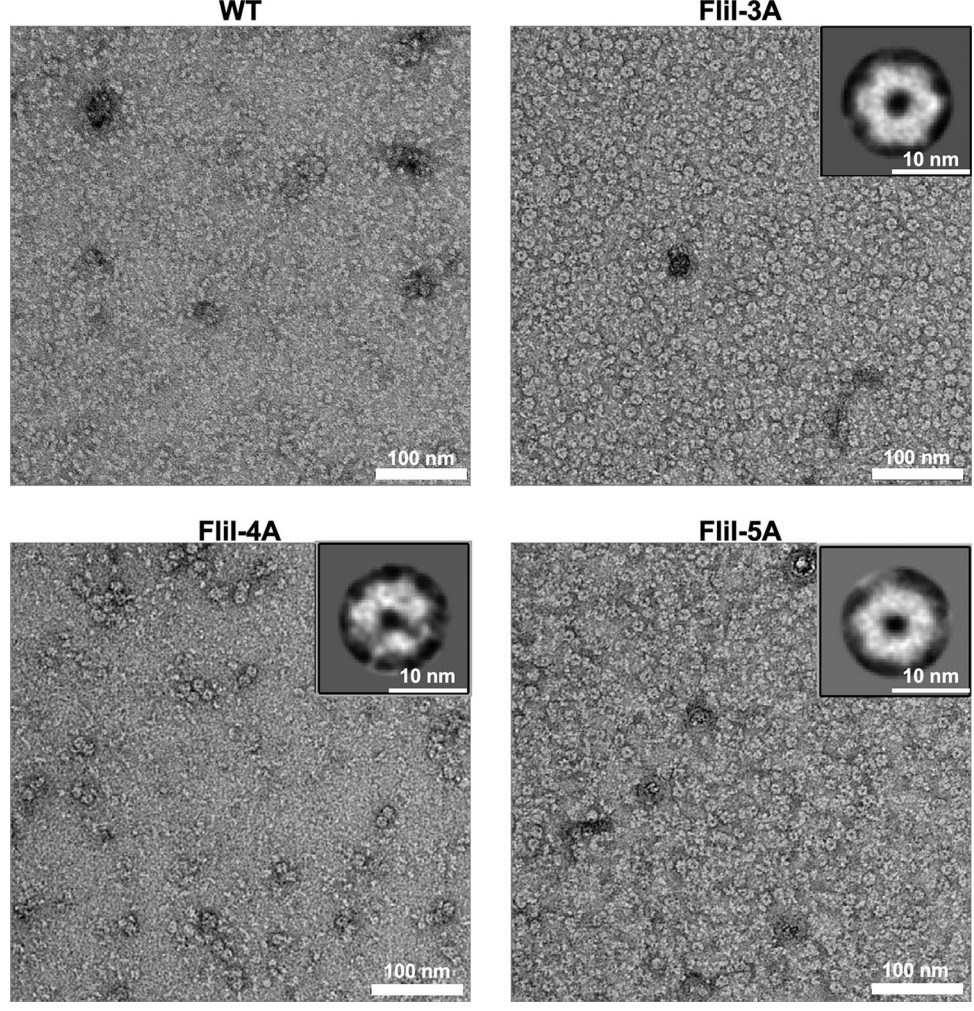

**Fig. 5 Electron micrographs showing the oligomerization ability of FliI, FliI-3A, FliI-4A, and FliI-5A.** Each purified FliI sample (1 μM) was preincubated with 5 mM MgCl$_2$, 5 mM ADP, 5 mM AlCl$_3$, and 15 mM NaF for 20 min at room temperature, and then the mixture was stained with 1% uranyl acetate and visualized by electron microscopy. Electron micrographs were recorded at a magnification of ×50,000. Insets indicate reference-free 2D class average images calculated by RELION3.0.7.

**Effect of the *fliI-5A* mutation on the ATPase activity of FliI.** FliI$_6$ ring formation is required for efficient flagellar protein export by the PMF-driven export gate complex even in the absence of FliH[19], raising the possibility that the *fliI-5A* mutation might suppress FliI oligomerization at the FlhA$_C$-FlhB$_C$ docking platform. To clarify this possibility, we measured the ATPase activity of FliI-5A at various protein concentrations (Fig. 4c and Table 1). The ATPase activity of FliI-5A was higher than the

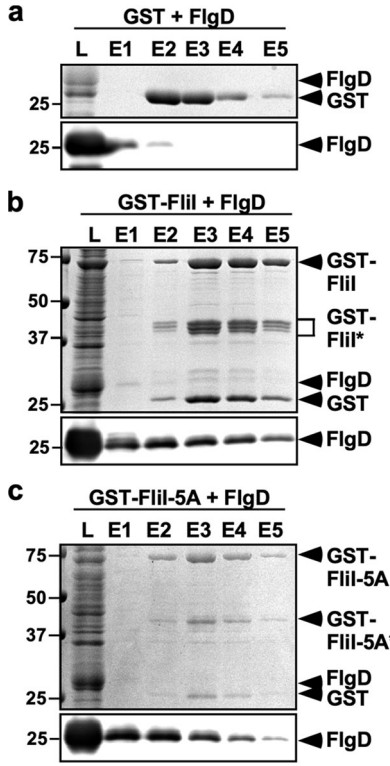

**Fig. 6 Effect of the *fliI-5A* mutation on the interaction of FliI with FlgD.** Cell lysates (indicated as L) prepared from *Salmonella* SJW1368 (Δ*cheW-flhD*) cells expressing **a** GST, **b** GST-FliI, or **c** GST-FliI-5A were mixed with those from *E. coli* BL21 (DE3) cells producing FlgD, and then each mixture was loaded onto a GST column. After washing with 15 ml of PBS, proteins were eluted with 5 ml of 50 mM Tris-HCl, pH 8.0, and 10 mM reduced glutathione. Elution fractions were analyzed by CBB staining (upper panel) and immunoblotting with anti-FlgD antibody (lower panel). The regions of interest were cropped from original CBB-stained gels and immunoblots shown in Supplementary Fig. 11. Three independent assays were performed.

wild-type value at the same protein concentrations. Consistently, FliI-5A formed the ring-shaped hexamer at a protein concentration of 1 μM (Fig. 5). Therefore, we hypothesize that the *fliI-5A* mutation may affect the gating function of the PMF-driven export gate complex in the absence of FliH.

**Isolation of pseudorevertants from the Δ*fliH fliI-5A* mutant.** To clarify our hypothesis described above, we isolated three pseudorevertants from the Δ*fliH fliI-5A* mutant. Motility of these pseudorevertants was better than that of their parental mutant (Fig. 8a). DNA sequencing revealed two insertion mutations, *flhB_{SP2}* and *flhB_{SP3}*, at the end of the C-terminal α-helix of FlhB_{CC} and a P361A missense mutation, named *flhB_{SP4}*, in the flexible C-terminal tail of FlhB_C (FlhB_{CCT}) (Fig. 8b). The inserted LKRWQ and WQLKR sequences of the Δ*fliH fliI-5A flhB_{SP2}* and Δ*fliH fliI-5A flhB_{SP3}* suppressor mutants, which are presumably caused by gene duplication, are located between Glu-349 and Leu-350 and between Arg-352 and Trp-353, respectively.

To test whether the *flhB_{SP2}*, *flhB_{SP3}* and *flhB_{SP4}* mutations display allele specificity, we transduced these *flhB* alleles into the Δ*fliH-fliI* mutant using P22 phage to produce the Δ*fliH-fliI flhB_{SP2}*, Δ*fliH-fliI flhB_{SP3}*, and Δ*fliH-fliI flhB_{SP4}* mutants. Motility of these three mutants was better than that of the Δ*fliH-fliI* mutant (Supplementary Fig. 5a). Consistently, much larger amounts of FlgD were detected in the culture supernatants of the Δ*fliH-fliI flhB_{SP2}*, Δ*fliH-fliI flhB_{SP3}*, and Δ*fliH-fliI flhB_{SP4}*

mutants than in that of the Δ*fliH-fliI* mutant (Supplementary Fig. 5b). These results indicate that these *flhB* mutations are able to bypass both FliH and FliI defects. Overexpression of wild-type FliI enhanced the motility of the Δ*fliH-fliI flhB_{SP4}* mutant (Fig. 8c). Consistently, FliI overexpression increased the secretion level of FlgD (Fig. 8d). In contrast, overexpression of FliI-5A reduced motility of and the secretion level of FlgD by the Δ*fliH-fliI flhB_{SP4}* mutant (Fig. 8d), indicating that FliI-5A still exerts an inhibitory effect on flagellar protein export even in the presence of *flhB_{SP4}* mutation. Similar results were obtained with the Δ*fliH-fliI flhB_{SP2}* and the Δ*fliH-fliI flhB_{SP3}* mutants (Supplementary Fig. 6). The second-site *flhB* mutations by themselves displayed no phenotype (Supplementary Fig. 5c, d). Because FliI-5A was fully functional at a relatively high copy level in the presence of FliH but not in its absence (Fig. 3), we suggest that the docking of FliI-5A to FlhB_C inhibits substrate entry into the polypeptide channel in the absence of FliH.

**Effect of *flhB* mutations on the interaction between FlhB_C and FlgD.** The N-terminal export signal of FlgD binds to a conserved hydrophobic patch formed by Ala-286, Pro-287, Ala-341, and Leu-344 residues of FlhB_C[36]. To test whether the *flhB* bypass mutations affect the interaction of FlhB_C with FlgD, we carried out pull-down assays by GST affinity chromatography. FlgD co-purified with GST-FlhB_C (Fig. 8e, first row). The *flhB_{SP2}*, *flhB_{SP3}*, and *flhB_{SP4}* mutations did not reduce the binding affinity of FlhB_C for FlgD (Fig. 8e), suggesting that these *flhB* mutations increase the probability of substrate entry into the FliP_5FliQ_4FliR_1 polypeptide channel complex and that FlhB_{CCT} may play a regulatory role in the substrate entry mechanism.

**Membrane topology of FlhB and FlhA.** The *flhB(P13T)*, *flhB (A21T/V)*, *flhB(I27N)*, and *flhB(P28T)* mutations, which are postulated to be located in the N-terminal cytoplasmic tail of FlhB (FlhB_{NCT}) (Fig. 8b), also bypass both FliH and FliI defects[34,37]. Recent genetic analysis has suggested that an interaction between FlhB_{NCT} and FlhB_{CN} may facilitate the entry of export substrates into the FliP_5FliQ_4FliR_1 polypeptide channel complex[38]. However, it remains unclear whether FlhB_{NCT} and FlhB_C are close to the cytoplasmic entrance of the polypeptide channel because the densities corresponding to these two parts of FlhB in the cryo-electron microscopic (cryoEM) structure of the core of the type III export apparatus were not high enough to build their atomic models[9]. Therefore, we analyzed the topology of FlhB_{TM} by PhoA fusion assays using L-broth agar plate containing a chromogenic substrate of alkaline phosphatase, 5-bromo-4-chloro-3-indolyl phosphate (BCIP). We constructed five FlhB–PhoA fusions, PhoA–FlhB, FlhB_{(1–59)}–PhoA, FlhB_{(1–132)}–PhoA, FlhB_{(1–184)}–PhoA, and FlhB_{(1–212)}–PhoA (Fig. 9a). We used a precursor form of PhoA (prePhoA) and its mature form (mPhoA) as the positive and negative controls, respectively, because prePhoA contains a signal peptide, allowing PhoA to be secreted via the Sec translocon into the periplasm to become the active form, whereas mPhoA does not. We also used a PhoA-FliQ fusion as a positive control because the cryoEM structure of FliQ has shown that the N-terminus of FliQ is located in the periplasm[6]. We used a *Salmonella* Δ*phoN* strain as a host. When prePhoA and the PhoA-FliQ fusion were expressed in the Δ*phoN* cells, the colonies became blue on the BCIP indicator plates (Supplementary Fig. 7). In contrast, the Δ*phoN* colonies expressing mPhoA remained white (Supplementary Fig. 7). These results indicate that the fusion of FliQ to the C-terminus of PhoA does not affect the PhoA activity. The Δ*phoN* cells expressing FlhB_{(1–59)}–PhoA or FlhB_{(1–184)}–PhoA formed blue colonies on

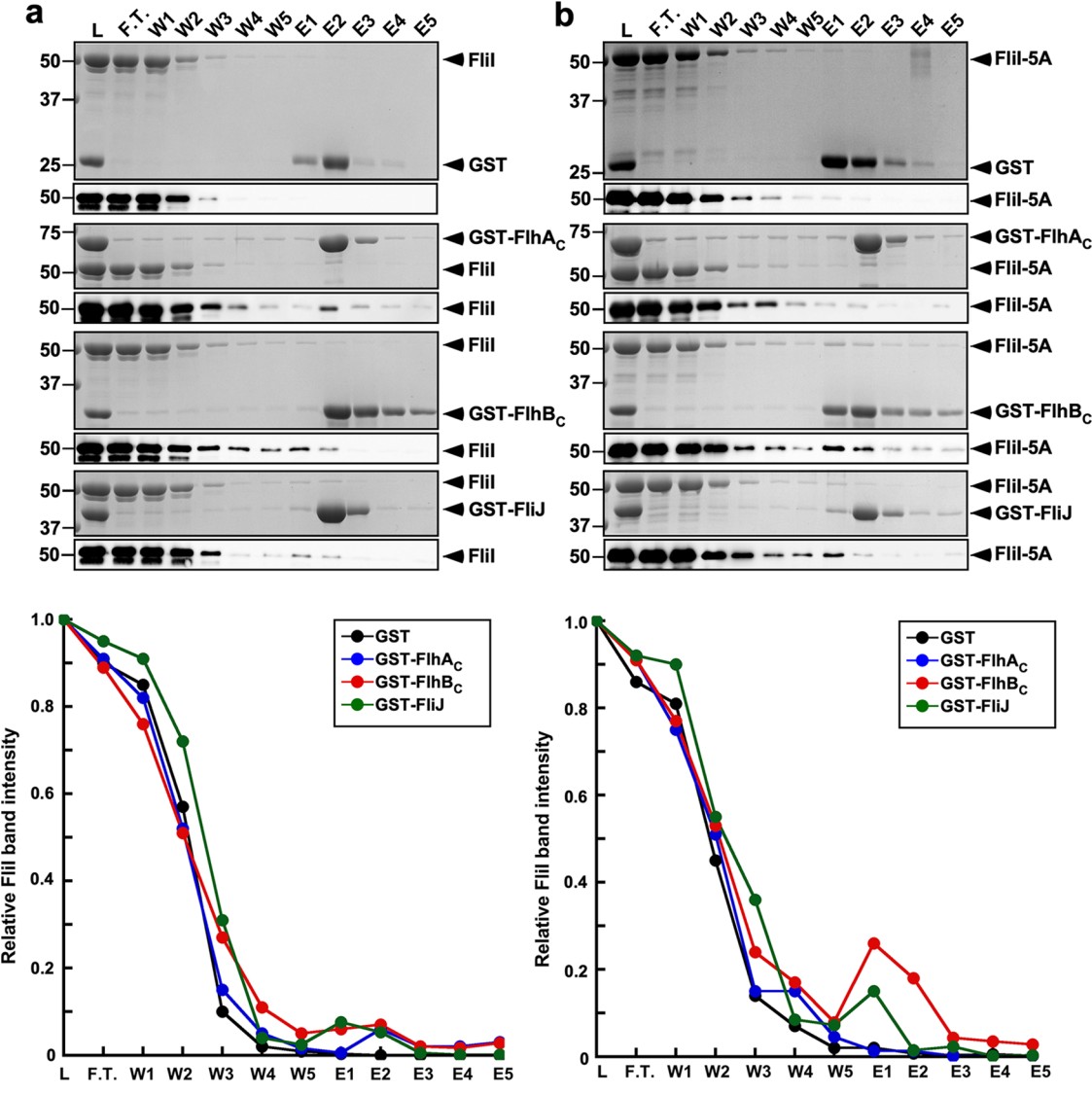

**Fig. 7 Effect of the *fliI-5A* mutation on the interactions of FliI with FlhA$_C$, FlhB$_C$, and FliJ. a** Purified His-FliI or **b** His-FliI-5A was mixed with GST-FlhA$_C$, GST-FlhB$_C$, or GST-FliJ, and then each mixture (L) was dialyzed overnight against PBS, followed by GST affinity chromatography. Flow through fraction (F. T.), wash fractions (W), and elution fractions (E) were analyzed by CBB staining (upper panel) and immunoblotting with anti-FliI antibody (lower panel). The regions of interest were cropped from original CBB-stained gels and immunoblots shown in Supplementary Fig. 12. The average band intensity of FliI in the L lane was set to 1.0, and then relative band intensity of each lane was calculated. These data are average of three independent experiments. Standard deviations were within 15%.

the BCIP indicator plates, whereas the cells expressing FlhB$_{(1-132)}$−PhoA formed white colonies (Fig. 9a), in agreement with the cryoEM structure of FlhB[9]. The PhoA−FlhB and FlhB$_{(1-212)}$−PhoA fusions showed no PhoA phosphatase activity (Fig. 9a), indicating that both FlhB$_{NCT}$ and FlhB$_C$ are located in the cytoplasm. Therefore, we suggest that they are in close proximity to each other in the export gate complex.

The *flhA(I21T)*, *flhA(L22F)*, and *flhA(V404M)* mutations have been identified as gain-of-function mutations that are able to overcome both FliH and FliI defects[34,37]. The *flhA(I21T)* and *flhA (L22F)* mutations lie in the N-terminal cytoplasmic tail of FlhA (FlhA$_{NCT}$), whereas the *flhA(V404M)* mutation is located in FlhA$_C$ (Fig. 9b). The FlhA$_C$ ring structure has been visualized to project into the cytoplasmic cavity of the C ring (Fig. 1)[10], but structural information of FlhA$_{TM}$ containing putative eight TM helices is lacking. Therefore, we constructed nine PhoA fusions, PhoA−FlhA, FlhA$_{(1-44)}$−PhoA, FlhA$_{(1-65)}$−PhoA, FlhA$_{(1-93)}$−PhoA, FlhA$_{(1-196)}$− PhoA, FlhA$_{(1-236)}$−PhoA, FlhA$_{(1-278)}$−PhoA, FlhA$_{(1-306)}$−PhoA, and

FlhA$_{(1-339)}$−PhoA (Fig. 9b). The FlhA$_{(1-44)}$−PhoA, FlhA$_{(1-93)}$−PhoA, FlhA$_{(1-236)}$−PhoA, and FlhA$_{(1-306)}$−PhoA fusions exhibited the phosphatase activity. However, the PhoA−FlhA, FlhA$_{(1-65)}$−PhoA, FlhA$_{(1-196)}$−PhoA, FlhA$_{(1-278)}$−PhoA, and FlhA$_{(1-339)}$−PhoA showed no phosphatase activity. These results are consistent with the membrane topology of FlhA predicted based on the primary sequence of FlhA. Because FlhA$_{NCT}$ was in the cytoplasm (Fig. 9b), we suggest that FlhA$_{NCT}$ is presumably close to FlhA$_C$, FlhB$_{NCT}$, and FlhB$_C$.

## Discussion
FliI forms a homo-hexamer at the flagellar base and hydrolyzes ATP to activate the PMF-driven export gate complex to drive flagellar protein export in a PMF-dependent manner (Fig. 1)[24]. FliI shows an extensive structural similarity with the α and β subunits of F$_1$-ATPase and hydrolyzes ATP with the mechanism similar to that of F$_1$-ATPase[18]. The α$_3$β$_3$ hetero-hexamer of F$_1$-ATPase is stabilized by interactions between the N-terminal

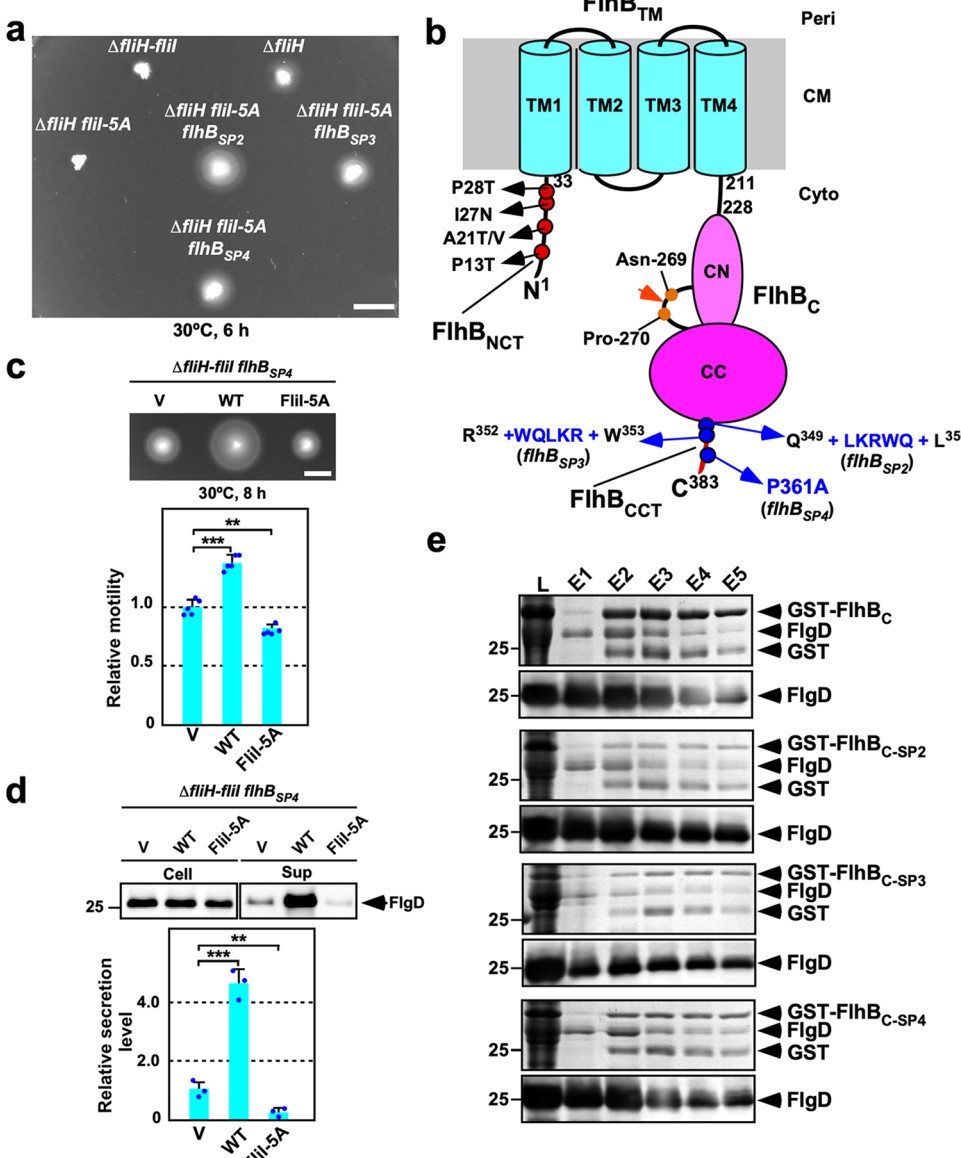

**Fig. 8 Isolation of pseudorevertants from the ΔfliH fliI-5A mutant. a** Motility of MMHI001 (Δ*fliH-fliI*) cells transformed with pTrc99A (indicated as Δ*fliH-fliI*), pMM1702 (indicated as Δ*fliH*), or pMM1702-5A (indicated as Δ*fliH fliI-5A*) and MMHI001-5A-SP2 transformed with pMM1702-5A (indicated as Δ*fliH fliI-5A flhB_SP2*), MMHI001-5A-SP3 transformed with pMM1702-5A (indicated as Δ*fliH fliI-5A flhB_SP3*), and MMHI001-5A-SP4 transformed with pMM1702-5A (indicated as Δ*fliH fliI-5A flhB_SP4*) in soft agar. The plate was incubated at 30 °C for 6 h. Scale bar, 0.5 cm. **b** Location of gain-of-function mutations in FlhB_C. FlhB_C undergoes autocatalytic cleavage between Asn-269 and Pro-270 residues to generate two distinct FlhB_CN (CN) and FlhB_CC (CC) polypeptides. FlhB_C has a highly flexible C-terminal cytoplasmic tail (FlhB_CCT). All gain-of-function mutations identified in this study are located in FlhB_CCT. The *flhB(P13T)*, *flhB(A21V)*, *flhB(A21T)*, *flhB(I27N)*, and *flhB(P28T)* mutations in the N-terminal cytoplasmic tail of FlhB (FlhB_NCT) have been identified as gain-of-function mutations that overcome the FliH and FliI defects to a considerable degree. **c** Motility of the Δ*fliH–fliI flhB_sp4* mutant carrying with pTrc99A (V), pMM1702 (WT), or pMM1702-5A (FliI-5A) in soft agar. The plate was incubated at 30 °C for 8 h. The diameter of the motility ring of five colonies of each strain was measured. The average diameter of the motility ring of the vector control was set to 1.0, and then relative diameter of the motility ring of each transformant was calculated. Dots indicate individual data points. Vertical bars indicate standard deviations. Scale bar, 0.5 cm. **d** Immunoblotting, using polyclonal anti-FlgD antibody, of whole-cell proteins and culture supernatant fractions prepared from the same transformants. The regions of interest were cropped from original immunoblots shown in Supplementary Fig. 13. Relative secretion levels of FlgD were measured. These data are average of three independent experiments. Dots indicate individual data points. Vertical bars indicate standard deviations. Comparisons between datasets were performed using a two-tailed Student's *t* test. A *P* value of <0.05 was considered to be statistically significant difference. \*\**P* < 0.01; \*\*\**P* < 0.001; ND, no statistical difference. **e** Effect of *flhB* mutations on the interaction of FlhB_C with FlgD. Whole-cell lysates (L) prepared from *Salmonella* SJW1368 (Δ*cheW-flhD*) cells expressing GST-FlhB_C, GST-FlhB_C-SP2, GST-FlhB_C-SP3, or GST-FlhB_C-SP4 were mixed with those from *E. coli* BL21 (DE3) cells producing FlgD, followed by GST affinity chromatography. Elution fractions were analyzed by CBB staining (upper panel) and immunoblotting with anti-FlgD antibody (lower panel). The regions of interest were cropped from original CBB-stained gels and immunoblots shown in Supplementary Fig. 14. Three independent assays were performed.

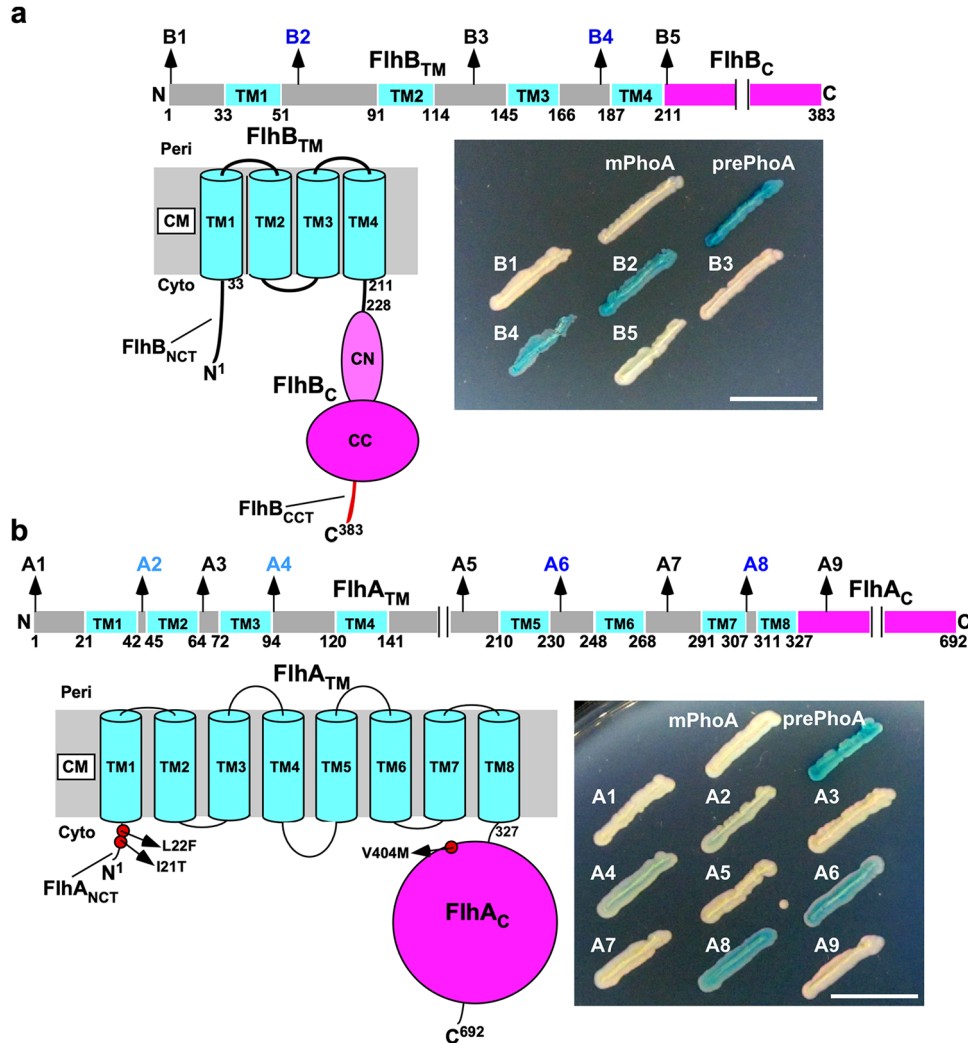

**Fig. 9 PhoA fusion assays. a** Membrane topology of FlhB. *Salmonella* TH12991 (Δ*phoN*) cells were transformed with pMKM10001 (pTrc99A/mPhoA), pMKM10002 (pTrc99A/prePhoA), pMKM10004 (pTrc99A/PhoA-FlhB, indicated as B1), pMKM10005 (pTrc99A/ FlhB(1–59)–PhoA, indicated as B2), pMKM10006 (pTrc99A/FlhB(1–132)–PhoA, indicated as B3), pMKM10007 (pTrc99A/FlhB(1–184)–PhoA, indicated as B4), or pMKM10008 pTrc99A/ FlhB(1–212)–PhoA, indicated as B5), and then fresh transformants were inoculated onto BCIP indicator plates containing 100 μg ml⁻¹ ampicillin, followed by incubation at 30 °C for 18 h. In the periplasm, PhoA adopts an active conformation to hydrolyze BCIP, thereby generating blue color colonies on BCIP indicator plates. In contrast, PhoA in the cytoplasm adopts an inactive form and hence cannot hydrolyze BCIP. As a result, the colonies remain white. Scale bar, 1.0 cm. **b** Membrane topology of FlhA. FlhA consists of the N-terminal transmembrane domain with eight putative transmembrane helices (TM1–TM8) (FlhA_TM) and the C-terminal cytoplasmic domain (FlhA_C). The *flhA(I21T)* and *flhA(L22F)* mutations in the N-terminal cytoplasmic tail of FlhA (FlhA_NCT) and the *flhA(V404M)* mutation in FlhA_C have been identified as gain-of-function mutations to partially rescue motility of the Δ*fliH* and Δ*fliH-fliI* mutant cells. Fresh TH12991 cells carrying pMKM10001, pMKM10002, pMKM10009 (pTrc99A/PhoA-FlhA, indicated as A1), pMKM10010 (pTrc99A/ FlhA(1–44)–PhoA, indicated as A2), pMKM10011 (pTrc99A/FlhA(1–65)–PhoA, indicated as A3), pMKM10012 (pTrc99A/FlhA(1–93)–PhoA, indicated as A4), pMKM10013 (pTrc99A/FlhA(1–196)–PhoA, indicated as A5), pMKM10014 (FlhA(1–236)–PhoA, indicated as A6), pMKM10015 (pTrc99A/FlhA(1–278)–PhoA, indicated as A7), pMKM10016 pTrc99A/FlhA(1–306)–PhoA, indicated as A8), or pMKM10017 pTrc99A/FlhA(1–339)–PhoA, indicated as A9) were inoculated onto BCIP indicator plates, and the plates were incubated at 30 °C for 18 h. Scale bar, 1.0 cm.

domains of the α and β subunits[39]. The core structure of FliI_N can be superimposed relatively well onto the N-terminal domains of the α and β subunits[18]. However, the FliI_6 ring model, which is generated by fitting the crystal structure of FliI onto the α and β subunits of the F_1-ATPase, shows steric hindrances of FliI_N at the subunit interface, suggesting that a conformational change of FliI_N is required for FliI ring formation[32]. Here, we showed that the *fliI(R33A)* and *fliI-3A* (R26A/R27A/R33A triple mutation) mutations increased the probability of FliI_6 ring formation at a relatively low protein concentration compared to wild-type FliI and that the *fliI(R76A)* mutation partially reduced the FliI ring formation efficiency of

FliI-3A (Figs. 4 and 5 and Table 1). These suggest that Arg-33 and Arg-76 are involved in well-regulated FliI ring formation (Fig. 2). The *fliI-3A* mutation reduced the binding affinity of FliH for FliI (Supplementary Fig. 2). However, when FliH was co-expressed with FliI-3A, it suppressed the growth defect caused by FliI-3A (Supplementary Fig. 3). Because FliH_C also binds to FliI_EN, we suggest that the interaction between FliH_C and FliI_EN suppresses premature FliI ring formation in the cytoplasm. Therefore, we propose that FliI requires FliH for well-regulated conformational rearrangements of FliI_N to form a homo-hexamer at the flagellar base and that Arg-33 and Arg-76 of FliI play important roles in the hexamer assembly.

FliI is not essential for flagellar protein export[37,40,41], and the FliI defect can be overcome to a considerable degree by many factors, such as bypass mutations in FlhA and FlhB, an increase in the expression level of export substrates, and an increase in total PMF[37,41]. Recently, in vitro reconstitution experiments using inverted membrane vesicles have demonstrated that the $FliH_2FliI$ complex facilitates the docking of export substrates and flagellar chaperone/substrate complexes to the transmembrane export gate complex so that the gate complex can efficiently unfold and transport export substrates in a PMF-dependent manner[30,42]. Based on these observations, FliI is thought to be involved in at least three distinct steps. First, FliI recognizes export substrates in the cytoplasm and delivers them to the $FlhA_C$-$FlhB_C$ docking platform along with FliH[26–29,31]. Second, ATP hydrolysis by FliI activates the export gate complex to become an active PMF-driven protein transporter[23,24]. Finally, FliI promotes export substrate entry into the $FliP_5FliQ_4FliR_1$ polypeptide channel complex[37]. Here, we showed that the $fliI$-5A (R26A/R27A/R33A/R76A/R93A) mutation caused a loss-of-function phenotype in the absence of FliH but not in its presence (Fig. 3). Interestingly, the level of FlgD secreted by the $\Delta fliH$ $fliI$-5A mutant was lower than that by the $\Delta fliH$-$fliI$ double null mutant (Fig. 3d). Furthermore, overexpression of FliI-5A reduced the secretion level of FlgD by the $\Delta fliH$-$fliI$ $flhB_{SP4}$ mutant compared to the vector control (Fig. 8d). These observations indicate that FliI-5A exerts an inhibitory effect on flagellar protein export in the absence of FliH. The binding affinities of FliI-5A for $FlhB_C$ and FliJ were higher than that of wild-type FliI (Fig. 7). In contrast, the binding affinity of FliI-5A for $FlhA_C$ was lower than that of wild-type FliI (Fig. 7). Because FliI-5A retained the ability to bind to FlgD (Fig. 6c), we suggest that the binding of FliI-5A to $FlhB_C$ and FliJ inhibits the flagellar protein export process in the absence of FliH. Therefore, we propose that a positive change cluster of FliI may regulate the binding affinities of FliI for $FlhA_C$, $FlhB_C$, and FliJ to facilitate the subsequent entry of flagellar building blocks into the $FliP_5FliQ_4FliR_1$ polypeptide channel complex and that the dissociation of FliI from $FlhB_C$ and FliJ may be required for efficient substrate entry into the polypeptide channel. Because FliI-5A was functional in the presence of FliH but not in its absence (Fig. 3), we also propose that an interaction between $FliH_C$ and $FliI_{EN}$ may allow the positive charge cluster of FliI to undergo its proper conformational changes coupled with the substrate entry process.

Gain-of-function mutations in $FlhB_{CCT}$ enhanced motility of the $\Delta fliH$ $fliI$-5A mutant cells (Fig. 8a). These $flhB_{CCT}$ mutations did not affect the binding affinity of $FlhB_C$ for FlgD (Fig. 8e). Furthermore, they did not display allele specificity (Supplementary Fig. 5c, d). These suggest that these $flhB_{CCT}$ mutations considerably increase the probability of export substrate entry into the $FliP_5FliQ_4FliR_1$ polypeptide channel complex in the absence of FliH and FliI. Therefore, we propose that $FlhB_{CCT}$ may suppress spontaneous gate opening of the polypeptide channel to avoid undesirable leakage of small salts through the channel. How does $FlhB_{CCT}$ control gate opening? $FlhB_{TM}$ associates with the $FliP_5FliQ_4FliR_1$ complex, and the cytoplasmic loop connecting helices 2 and 3 ($FlhB_{Loop}$) wraps around the cytoplasmic face of the $FliP_5FliQ_4FliR_1$ complex through interactions of $FlhB_{Loop}$ with each FliQ subunit, having proposed that $FlhB_{Loop}$ may be involved in the gating mechanism of the polypeptide channel[9]. The $flhB(E230A)$ mutation in $FlhB_{CN}$ reduces the protein transport activity of the flagellar protein export apparatus, and the $flhB(E11S)$ mutation in $FlhB_{NCT}$ restores the protein transport activity of the $flhB(E230A)$ mutant to the wild-type level[38], suggesting that an interaction between $FlhB_{NCT}$ and $FlhB_{CN}$ is required for flagellar protein export. We found that the $flhB(E230A)$ mutation did not reduce the binding affinities of $FlhB_C$ for either FliI or FlgD (Supplementary Fig. 8). PhoA fusion assays showed that

both $FlhB_{NCT}$ and $FlhB_C$ are in the cytoplasm (Fig. 9a), suggesting that they are close to $FlhB_{Loop}$. Recent photo-crosslinking experiments have shown that Pro-28 of $FlhB_{NCT}$ is in very close proximity to FliQ[9]. These suggest that conformational rearrangements of $FlhB_{NCT}$, $FlhB_{Loop}$, and $FlhB_{CN}$ occur in a FliI-dependent manner, allowing the polypeptide channel to be opened and that $FlhB_{CCT}$ may regulate the $FlhB_{NCT}$–$FlhB_{CN}$ interaction to interfere with premature gate opening.

The $flhB(P13T)$, $flhB(A21V)$, $flhB(A21T)$, $flhB(I27N)$, and $flhB(P28T)$ mutations in $FlhB_{NCT}$; the $flhA(I21T)$, and $flhA(L22F)$ mutations in $FlhA_{NCT}$; and the $flhA(V404M)$ mutation in $FlhA_C$ overcome both FliH and FliI defects[34,37]. PhoA fusion assays revealed that $FlhA_{NCT}$ is located in the cytoplasm (Fig. 9b), suggesting that $FlhA_{NCT}$ is in close proximity to $FlhA_C$, $FlhB_{NCT}$, and $FlhB_C$. Therefore, we propose that gain-of-function mutations in FlhA and FlhB may mimic FliI-bound state of the cytoplasmic entrance of the polypeptide channel so that an interaction between flagellar building block and $FlhB_C$ may trigger the gate opening in the absence of FliH and FliI.

## Methods

**Bacterial strains, plasmids, transductional crosses, and DNA manipulations**. Bacterial strains and plasmids used in this study are listed in Supplementary Table 1. P22-mediated transductional crosses were performed with P22HTint. Site-directed mutagenesis was carried out using the QuikChange Site-Directed Mutagenesis method as described in the manufacturer's instructions (Stratagene). DNA sequencing reactions were carried out using BigDye v3.1 (Applied Biosystems), and then the reaction mixtures were analyzed by a 3130 Genetic Analyzer (Applied Biosystems).

**Motility assays in soft agar**. Fresh colonies were inoculated into soft agar plates [1% (w/v) triptone, 0.5% (w/v) NaCl, 0.35% Bacto agar] with or without 100 µg ml$^{-1}$ ampicillin and incubated at 30 °C. A diameter of the motility ring was measured using the ImageJ software version 1.52 (National Institutes of Health). At least five different colonies were measured.

**Secretion assays**. S. enterica wild-type and mutant cells were grown at 30 °C with shaking until the cell density had reached an optical density of 600 nm ($OD_{600}$) of ca. 1.4–1.6. Cultures were centrifuged to obtain cell pellets and culture supernatants. The cell pellets were resuspended in a sample buffer solution [62.5 mM Tris-HCl, pH 6.8, 2% sodium dodecyl sulfate (SDS), 10% glycerol, 0.001% bromophenol blue] containing 1 µl of 2-mercaptoethanol. Proteins in the culture supernatants were precipitated by 10% trichloroacetic acid and suspended in a Tris/SDS loading buffer (one volume of 1 M Tris, nine volumes of 1× sample buffer solution) containing 1 µl of 2-mercaptoethanol. Both whole cellular proteins and culture supernatants were normalized to a cell density of each culture to give a constant cell number. After boiling proteins in both whole cellular and culture supernatant fractions at 95 °C for 3 min, these protein samples were separated by SDS–polyacrylamide gel (normally 12.5% acrylamide) electrophoresis (SDS–PAGE) and transferred to nitrocellulose membranes (Bio-Rad) using a transblotting apparatus (Hoefer). Then immunoblotting with polyclonal anti-FlgD antibody was carried using iBand Flex Western Device (Thermo Fisher Scientific) as described in the manufacturer's instructions. Detection was performed with Amersham ECL Prime western blotting detection reagent (Cytiva). Chemiluminescence signals were captured by a Luminoimage analyzer LAS-3000 (GE Healthcare). The band intensity of each blot was analyzed using an image analysis software, CS Analyzer 4 (ATTO, Tokyo, Japan). All image data were processed with the Photoshop software (Adobe). At least three independent measurements were performed.

**Pull-down assays by Ni affinity chromatography**. The S. enterica SJW1368 strain carrying pTrc99A-based plasmids co-expressing of untagged FliH with His-FliI or its mutant variants were grown overnight at 30 °C in 100 ml L-broth containing ampicillin. Cell lysates were loaded onto a nickel-nitriloacetic acid (Ni-NTA) agarose column (QIAGEN). After washing the column with 20 mM Tris-HCl, pH 8.0, 500 mM NaCl, and 50 mM imidazole, proteins were eluted with 20 mM Tris-HCl, pH 8.0, and 500 mM NaCl containing imidazole by a stepwise increase in the imidazole concentration of 100, 250, and 500 mM. Fractions were analyzed by SDS-PAGE with Coomassie Brilliant blue (CBB).

**Cell growth measurements**. Overnight cultures of S. enterica cells were diluted 100-fold into fresh L-broth containing 100 µg ml$^{-1}$ ampicillin, and the cells were grown at 30 °C for 3 h with shaking. After adding IPTG at a final concentration of 0.1 mM, the incubation was continued for another 4 h. The cell growth was

monitored at an $OD_{600}$ every hour. Three different cells were measured and averaged.

**Purification of wild-type and point mutant variants of FliI.** *Escherichia coli* BL21 (DE3) Star cells carrying an appropriate pET19b-based plasmid encoding His-FliI, His-FliI(R33A), or His-FliI-3A were grown overnight at 30 °C in 250 ml of L-broth containing 100 μg ml$^{-1}$ ampicillin. His-FliI and its mutant variants were purified from cell lysates by Ni affinity chromatography with a Ni-NTA agarose column (QIAGEN), followed by size exclusion chromatography with a Hi-Load Superdex 200 (26/60) column (GE Healthcare)[43]. For purification of His-FliI-4A and His-FliI-5A, the *S. enterica* SJW1368 cells transformed with pMKM1702-4AiH or pMKM1702-5AiH, which encodes His-FliI-4A + FliH or His-FliI-5A + FliH on the pTrc99A vector, were grown overnight at 30 °C in 250 ml of L-broth containing ampicillin. His-FliI-4A/FliH and His-FliI-5A/FliH complexes were purified from the soluble fractions by Ni affinity chromatography, followed by size exclusion chromatography to remove FliH. Fractions containing His-FliI or its mutant variants were dialyzed overnight against 50 mM Tris-HCl, pH 8.0, 150 mM NaCl, and 1 mM EDTA at 4 °C.

**Measurements of the FliI ATPase activity.** Wild-type FliI, FliI(R33A), FliI-3A, FliI-4A, and FliI-5A were concentrated to 40, 5, 5, 20, and 20 μM, respectively, and then each purified sample was added to a buffer containing 30 mM HEPES-NaOH, pH 8.0, 30 mM KCl, 30 mM NH₄Cl, 5 mM Mg(CH₃COO)₂, 1 mM dithiothreitol (DTT), 0.5 mg ml$^{-1}$ bovine serum albumin (Sigma), and 4 mM ATP, followed by the incubation at 37 °C. The ATPase activities of purified FliI and its mutant variants were measured at different protein concentrations by the Malachite Green assay[44]. At least three measurements were carried out at each FliI protein concentration.

**In vitro reconstruction of the FliI ring structure.** Purified proteins (final concentration 1 μM) was incubated in 50 mM Tris–HCl, pH 8.0, 113 mM NaCl, 0.8 mM EDTA, 1 mM DTT, 5 mM MgCl₂, 5 mM ADP, 5 mM AlCl₃, and 15 mM NaF at room temperature for 20 min. Samples were applied to carbon-coated copper grids and negatively stained with 2% (w/v) uranyl acetate. Electron micrographs were recorded at a magnification of ×50,000 with a JEM-1011 transmission electron microscope (JEOL, Tokyo, Japan) operated at 100 kV. To carry out two-dimensional class averaging of the FliI-3A, FliI-4A, and FliI-5A ring structures, 154, 100, and 103 particle images were picked manually, aligned, classified, and averaged using the RELION3.0.7 program[45].

**Purification of GST-tagged proteins.** Cell lysates prepared from the *S. enterica* SJW1368 strain expressing GST-tagged proteins were loaded onto a Glutathione Sepharose 4B column (GE Healthcare). After washing with phosphate-buffered saline (PBS; 8 g of NaCl, 0.2 g of KCl, 3.63 g of Na₂HPO₄•12H₂O, 0.24 g of KH₂PO₄, pH 7.4 per liter), proteins were eluted with 50 mM Tris-HCl, pH 8.0, and 10 mM reduced glutathione. Fractions containing GST-tagged proteins were pooled and dialyzed overnight against PBS at 4 °C with three changes of PBS.

**Pull-down assays by GST affinity chromatography.** To investigate the effect of the *fliI-5A* mutation on interactions of FliI with FlhA$_C$, FlhB$_C$ and FliJ pull-down assays by GST affinity chromatography were carried out as described previously[46]. Purified His-FliI or His-FliI-5A was mixed with purified GST, GST-FlhA$_C$, GST-FlhB$_C$, or GST-FliJ, and then each mixture was dialyzed overnight against PBS at 4 °C with three changes of PBS. A 5 ml of each mixture was loaded onto a Glutathione Sepharose 4B column (bed volume, 1 ml) pre-equilibrated with 20 ml of PBS. After washing of the column with 10 ml PBS at a flow rate of ca. 0.5 ml min$^{-1}$, bound proteins were eluted with 50 mM Tris-HCl, pH 8.0, and 10 mM reduced glutathione. Fractions were analyzed by SDS-PAGE with CBB staining and immunoblotting with polyclonal anti-FliI antibody. The band intensity of each blot was analyzed using an image analysis software, CS Analyzer 4.

To analyze the interactions of FlgD with FliI and FlhB$_C$, cell lysates prepared from the *S. enterica* SJW1368 strain expressing GST, GST-FliI, GST-FliI-5A, GST-FlhB$_C$, GST-FlhB$_{C-SP2}$, GST-FlhB$_{C-SP3}$, GST-FlhB$_{C-SP4}$ or GST-FlhB$_{C-E230A}$ were mixed with those from the *E. coli* BL21 (DE3) Star strain overexpressing His-FlgD, and each mixture was loaded onto a Glutathione Sepharose 4B column. After extensive washing the column with 10 ml PBS, bound proteins were eluted with 50 mM Tris-HCl, pH 8.0, and 10 mM reduced glutathione. Eluted fractions were analyzed by CBB staining and immunoblotting with polyclonal anti-FlgD antibody.

**PhoA fusion assays.** Fresh transformants were inoculated onto BCIP indicator plates [1% (w/v) triptone, 0.5% (w/v) NaCl, 0.35% (w/v) Bacto agar, 50 μg ml$^{-1}$ BCIP] containing 100 μg ml$^{-1}$ ampicillin and incubated at 30 °C for 18 h. At least seven independent measurements were performed.

**Multiple sequence alignment.** Multiple sequence alignment was carried out using CLUSTAL-Ω (http://www.ebi.ac.uk/Tools/msa/clustalo/).

**Statistics and reproducibility.** Statistical tests, sample size, and the number of biological replicates are reported in the figure legends. Statistical analyses were done using the Excel software (Microsoft). Comparisons between datasets were performed using a two-tailed Student's *t* test. A *P* value of <0.05 was considered to be statistically significant difference. *$P < 0.05$; **$P < 0.01$; ***$P < 0.001$.

**Reporting summary.** Further information on research design is available in the Nature Research Reporting Summary linked to this article.

## Data availability
All data generated during this study are included in this published article and its Supplementary Information files. Strains, plasmids, polyclonal antibodies, and all other data are available from the corresponding author on reasonable request.

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

## Acknowledgements

We thank Kelly T. Hughes for his kind gift of *Salmonella phoN* mutant strains and Tomoko Miyata, Fumiaki Makino and Yasuyo Abe for technical assistance. This work was supported in part by the Japan Society for the Promotion of Science (JSPS KAKENHI Grant Numbers JP18K14638 and JP20K15749 to M.K., JP25000013 to K.N., and JP26293097 and JP19H03182 to T.M.) and JEOL YOKOGUSHI Research Alliance Laboratories of Osaka University to K.N.

## Author contributions

M.K., K.N., and T.M. designed research; M.K. and T.M. performed research; M.K. and T.M. analyzed data; M.K., K.N., and T.M. wrote the paper.

## Competing interests

The authors declare no competing interests.
