## [Peer Review File · Communications Biology]

Reviewers' comments:

Reviewer #1 (Remarks to the Author):

In the present manuscript, Kinoshita and colleagues investigate the contribution of the cytoplasmic ATPase complex made of FliH2I6J to protein export via the flagellar type-III secretion system (T3SS). The T3SS is intrinsically a proton motive force (PMF) powered protein export machine. The cytoplasmic ATPase complex FliH2I6J contributes to efficient protein export and ATPase hydrolysis is thought to activate the PMF-driven protein export process.

The ATPase FliI forms a homohexamer dependent on FliJ and its ATPase activity is regulated by FliH. The N-terminal domain of FliI is important for FliI hexamerization to exert ATPase activity. FliJ binds to the FlhA component of the T3SS and plays an important role for the ATP-dependent activation of the export gate. The C-terminal domain of FliH binds to FliI_N and a charged residue cluster at the interface of two FliI monomers, thereby inhibiting FliI hexamer formation and FliI ATPase activity. Accordingly, it is assumed that FliH plays a regulatory role during the flagellar protein export process.

However, the molecular mechanism how ATPase hydrolysis via the ATPase complex activates the PMF-driven export gate complex remains unclear. Here, Kinoshita et al. performed a mutational analysis to investigate the positively charged residue cluster of FliI and present a revised model of the energy coupling mechanism of the flagellar protein export apparatus.

The reported results and revised model are interesting to the field and will help to elucidate the molecular mechanisms of protein export via the T3SS. I have the following comments that might help the authors to improve the clarity of the manuscript and presentation of their data.

Main comments:

- Figure 2: How do plasmid expression levels of FliI compare to physiological FliI expression? I was surprised to see that the authors were not able to detect FliI when expressed from the low copy plasmid.
- Line 126-127: To me, it looks like the binding of FliH to FliI is also reduced for the mutations 2A, 3A and 4A. How do you explain that you do not observe the same phenotype in motility and secretion as for 5A?
- Line 153-155: it is unclear how the statement that the fliI-3A mutation significantly reduced the binding affinity of FliI for FliH agrees with the results reported in reference 19?
- Line 157-160: Do you see the same growth defects for 4A and 5A? That's what you would expect if reduced binding of FliH is responsible for premature FliI ring formation and thus ATP hydrolysis?
- Line 212-214: Why is FlgD secreted in the fliH-fliI mutant + vector control? Is this due to cell lysis?
- Line 214-215: Why are the FliI protein levels reduced in R33A and 3A?
- Line 216-217: The authors should highlight before that the R93 mutation has a decreased secretion of FlgD compared to the other single mutations. It might be of interest to determine secretion levels of late substrates as well, e.g. FliC.
- Line 218-223: The authors mention that chaperone-late substrate complexes bind to FliI, but then they investigate the binding of FlgD to FliI and see no difference. Perhaps it would have been useful to look at binding deficiencies of chaperone-late substrate complexes?
- Figure 4B: FliI must have been mislabeled. FliI is indicated as ~37 kDa in size in the blot of the second row (GST-FlhAC), but as 50 kDa in size in all the other blots?
- Discussion: I found it very difficult to follow all the different mutations in FlhA and FlhB and their suppressor mutations in various backgrounds and their effect on secretion from previously published papers. Perhaps, these descriptions can be simplified/summarized? I would also consider to place the observations on FlhA topology in the results and not solely in the discussion part.
- The proposed mechanism of the flagellar T3SS is quite speculative and I am not convinced the model is very well supported by the presented data. Certainly, additional data on the dynamic

interactions between FlhB/FlhA/FliH/FliI are needed to substantiate the model in future studies. For the present manuscript, my recommendation would be to discuss the open mechanistic questions in more detail. Just to name a few... what are potential functions of the ATPase complex after activation of the PMF-driven protein export? Does the escort function of FliH2-FliI really play a functional role (e.g. PMID 25068520)? In all the cryo electron tomography images, the ATPase complex appears to be rather stably associated to the base of the export gate. It might also worthwhile to note that protein export can actually occur in the absence of the ATPase complex (e.g. PMID 25393010, 31911488). What about substrate unfolding via the ATPase as frequently cited as a potential role (e.g. PMID 32346005, 16208377)?

Minor comments

- Line 26: ...mutations conferred a loss-of-function...
- Line 40-44: References missing
- Line 51-52: References missing (e.g. by Kuhlen et al. 2018)
- Line 61: Missing reference indicated as 14 (Fabiani et al.)
- Line 54-55: Transmembrane domain is misleading, as the proton coupling is thought to be mediated by a cytoplasmic loop between two transmembrane helices and the large cytoplasmic domain of FlhA
- Line 66: basal body; C-ring rather than C ring
- Line 65-68: Include other relevant references
- Line 83-86: Sentence is difficult to read.
- Line 109: Why did you exclude R76?
- Supplemental Fig. 1 and Figure 2A, Figure 3 (+ other Figures with motility plates): Did the plates contain supplements to induce the plasmids? According to the Methods section they don't, but I would suggest to include this information in the Figure legend.
- Supplemental Fig. 2: Indicate that I and H correspond to FliI and FliH
- Figure 4A: What is GST-FliI*?
- Line 231: mutation...FliI
- Figure 5: Include genotype to the corresponding strain number (e.g. MMHI1001 (Δ fliH-fliI)). It would be easier to follow.
- Line 281: Reference to Figures doesn't match. Figure 2 and 3 instead?
- Line 305: FlhB
- Line 309: ...to hydrolyze..
- Line 343: have shown
- Line 354: ...hetero-hexamer of the...
- Line 363-364: were over-expressed
- Line 395: Figure 5D?
- Line 396: FliHEN was not specified before.
- Line 397: interacts
- Line 428-432: Consider rephrasing to make it more easy to understand
- Line 461: Based on
- Line 468: ...thereby opening the polypeptide channel...
- Line 470: through
- Line 510: inoculated into

Reviewer #3 (Remarks to the Author):

The work by Kinoshita et al aims to analyze the role of positively charged residues at the interface of the flagella T3SS ATPase FliI and FliH in regulating T3S-dependent export.

The paper is well written and the figures are clear. Only the discussion is a bit lengthy (it is a review in itself, not very focussed) and the narrative jumps somewhat between different mutants.

General remarks and major points:

The authors show that the simultaneous mutation of five arginines (FliA-5A) impairs secretion through the flagellar T3SS and consequently flagellar motility while the mutation of fewer arginines does not make a difference. The secretion and motility defect of the FliI-5A mutant can be rescued by overexpression of the mutated protein.

FliI is thought to exist in a flagellum-bound hexameric form that shows high ATPase activity and in a flagellum-unbound FliI1-FliH2 complex that shows low ATPase activity. Therefore it was suggested that the binding of FliH to FliI prevents FliI ring formation and thus indirectly ATPase activity. The authors show that the FliI-5A mutation impairs the interaction of FliI with FliH and conclude that the impaired interaction causes the secretion and motility defect. Interestingly, also FliI-2A, FliI-3A and FliI-4A mutations that don't show a defect in motility show the same defect in interaction of FliI and FliH. Consequently, the suggested link from a defect in FliI-FliH interaction to a secretion defect is not conclusive.

Furthermore, FliI-5A overexpression cannot compensate for a fliH deletion as the FliI wild type and other mutants can, showing that FliI-5A also has a defect independent of the FliI-FliH interaction.

The authors test whether the FliI-5A defect results from an impaired binding to T3SS substrates, but this is not the case. They then sought to assess whether the FliI-5A mutant showed a reduced binding to the flagellar components FlhA, FlhB and FliJ. The authors claim that no difference in FliI-binding to these proteins is observable between the wild type and the mutant. However, there is no relevant binding observed between any of these proteins at all (maybe with the exception of an almost insignificant FliI-FlhA interaction), precluding the drawing of any conclusions.

The authors then isolated pseudorevertants of the secretion and motility-deficient Δ fliH, FliI-5A mutant that located to the very C-terminus of FlhB. These FlhB mutants substantially rescued the secretion of a Δ fliH, Δ fliI double mutant, suggesting to me that they increase the secretion capacity of the flagellar T3SS in a more general way.

Based on all these results the authors hypothesize that an interaction between FliI and FlhB triggers the opening of export gate formed by FliPQR and FlhB. Unfortunately, I can't see the conclusiveness of this hypothesis as the FlhB mutations act independently of FliI altogether. Furthermore, based on our current structural understanding, we have no indication whatsoever that FliI and FlhB might interact. They are likely to reside on different sides of the FlhAc nonamer, unable to interact. No one has shown up to this point that FliI and FlhB interact in an intact system in vivo. I am afraid that weak GST-pulldown data do not suffice to make this point.

The authors show further that overexpression of the mutants FliIR33A and FliI-3A imposes a growth defect on the bacteria that can be rescued by co-overexpression of FliH. Unfortunately, the assaying of the growth defect was done by a barely quantitative plating method and should be repeated with more quantitative assays. The authors postulate that the putative growth defect results from futile ATP hydrolysis caused by an increased ring-forming propensity of these FliI mutants, which is suppressed by FliH-binding. Indeed, an increased ATP hydrolysis was measured in these mutants, however, the data are not fully convincing as very different ATP concentrations were tested to assess the activity of the wild type and the mutant. To improve this dataset, the assays should be repeated at the same ATP concentrations to enable full comparison.

In this section, the authors speculate much on the effect of the tested mutations on FliI ring formation but it is never directly tested. It would be beneficial for the paper to show that FliI ring formation is truly effected, i.e., by size exclusion chromatography or crosslinking.

Overall, the presented work analyzes the effects of mutations in FliI and FlhB but unfortunately, the presented results do not add substantial insight to our current understanding of the T3SS function in general and export gate opening in particular. Thus, I feel that the major conclusions and claims of

this paper are overstated. In particular, there is no evidence presented that FliI and FlhBc interact to open the export gate, as is claimed by the authors.

Minor points:

Page 1 and 2: Title and abstract should be adjusted to reflect the results. Both are much overstated.

Page 4, line 12: "basal" body

Page 5, line 10: "chaperone"

Page 9, lines 3-10: Please draw a graph of all ATPase assay data.

Page 12, line 9: "mutation; "FIII" should be "FliI"

Page 15 ff: I don't see the point of this topology assay as the topology is obvious from the solved FlhB structure.

Page 17, line 17: "have" instead of "has"

Page 43: The model drawn is not correct with respect to the location of the FlpQR FlhB complex.

To Reviewer #1:

In the present manuscript, Kinoshita and colleagues investigate the contribution of the cytoplasmic ATPase complex made of FliH2I6J to protein export via the flagellar type-III secretion system (T3SS). The T3SS is intrinsically a proton motive force (PMF) powered protein export machine. The cytoplasmic ATPase complex FliH2I6J contributes to efficient protein export and ATPase hydrolysis is thought to activate the PMF-driven protein export process.

The ATPase FliI forms a homohexamer dependent on FliJ and its ATPase activity is regulated by FliH. The N-terminal domain of FliI is important for FliI hexamerization to exert ATPase activity. FliJ binds to the FlhA component of the T3SS and plays an important role for the ATP-dependent activation of the export gate. The C-terminal domain of FliH binds to FliI_N and a charged residue cluster at the interface of two FliI monomers, thereby inhibiting FliI hexamer formation and FliI ATPase activity. Accordingly, it is assumed that FliH plays a regulatory role during the flagellar protein export process.

However, the molecular mechanism how ATPase hydrolysis via the ATPase complex activates the PMF-driven export gate complex remains unclear. Here, Kinoshita et al. performed a mutational analysis to investigate the positively charged residue cluster of FliI and present a revised model of the energy coupling mechanism of the flagellar protein export apparatus.

The reported results and revised model are interesting to the field and will help to elucidate the molecular mechanisms of protein export via the T3SS. I have the following comments that might help the authors to improve the clarity of the manuscript and presentation of their data.

Re: Thank you very much for your supportive comments.

Main comments:

- Figure 2: How do plasmid expression levels of FliI compare to physiological FliI expression? I was surprised to see that the authors were not able to detect FliI when expressed from the low copy plasmid.*

Re: When FliI is expressed from a pET19b-based plasmid, its expression level is almost the same as the chromosomal expression level. In contrast, when FliI is expressed from a pTrc99A-based plasmid, the expression level increases by more than 100-fold. To see the expression levels of wild-type FliI and its mutant variants expressed from the pET19b-based plasmid, we prepared whole cell lysates from a *fliI* null mutant transformed with the pET19b-based plasmid and performed immunoblotting with polyclonal anti-FliI antibody. Now the FliI bands are clearly visualized on immunoblots (**New Supplementary Fig. 1b**).

- Line 126-127: To me, it looks like the binding of FliH to FliI is also reduced for the mutations 2A, 3A and 4A. How do you explain that you do not observe the same phenotype in motility and secretion as for 5A?*

Re: An interaction between the extreme N-terminal α -helix of FliI (FliI_{EN}) and FliH_C is critical for the formation of the FliH₂FliI complex in solution (**Minamino and Macnab. *Mol. Microbiol.* 37, 1494–1503. 2000**). FliI_{EN} forms not only hydrophobic but also electrostatic interaction networks with FliH_C (**Imada *et al.* *PNAS* 113, 3633–3638. 2000**). Because strong ion strength weakens the electrostatic interactions, we assume that FliH easily dissociates from FliI-2A, FliI-3A, FliI-4A and FliI-5A during Ni affinity chromatography in which the binding buffer contains 500 mM NaCl. FliI-2A, FliI-3A and FliI-4A fully restored the motility of the Δ *fliI* mutant when they were expressed from pET19b-based plasmids whereas FliI-5A did not. Since over-expression of FliI-5A restored motility to the wild-type level, we suppose that the binding affinity of FliI-5A for FliH_C is weaker than those of FliI-2A, FliI-3A and FliI-4A under physiological conditions.

• *Line 153-155: it is unclear how the statement that the fliI-3A mutation significantly reduced the binding affinity of FliI for FliH agrees with the results reported in reference 19?*

Re: Imada *et al.* (2016) have already reported that the *fliI*-3A mutation (R26A/R27A/R33A triple mutation) reduces the binding affinity of FliI for FliH as judged by pull down assays using Ni affinity chromatography [Fig. S5 in the 2016 PNAS paper by Imada *et al.* (***PNAS* 113, 3633–3638. 2016**)].

• *Line 157-160: Do you see the same growth defects for 4A and 5A? That's what you would expect if reduced binding of FliH is responsible for premature FliI ring formation and thus ATP hydrolysis?*

Re: Over-expression of FliI-4A and FliI-5A did not cause the growth arrest, similarly to wild-type FliI (**new Fig. 3a**). Co-expression of FliI-4A and FliI-5A with FliH increased the solubility of these two FliI mutant proteins (**new Fig. 3b and new Supplementary Fig. 3c**), suggesting that FliH binds to FliI_{EN} of these two mutant proteins. FliI-4A and FliI-5A formed a homo-hexamers but their ring formation efficiency was much lower than FliI-3A (**new Fig. 4**) and hence their ATPase activity was lower at a low protein concentration compared to FliI-3A (**new Fig. 3c and Table 1**). These observations suggest that the R76A mutation reduces the probability of premature ring formation of FliI-3A significantly, thereby suppressing the growth arrest caused by over-expression of FliI-3A.

• *Line 212-214: Why is FlgD secreted in the fliH-fliI mutant + vector control? Is this due to cell lysis?*

Re: We do not think that it is due to cell lysis. Because the *fliH-fliI* double null mutant displayed a weakly motile phenotype (**see new Supplementary Fig. 4a**), it is reasonable that we sometimes detect FlgD in the culture supernatants of this *fliH-fliI* double null mutant.

• *Line 214-215: Why are the FliI protein levels reduced in R33A and 3A?*

Re: When we prepared the pTrc99A-based plasmids encoding FliI with the R33A or FliI-3A mutation from *Salmonella* strains, the plasmid concentrations were reproducibly

lower than the wild-type FliI plasmid presumably because FliI(R33A) and FliI-3A waste ATP a lot to reduce the cytoplasmic ATP level enough to inhibit the cell growth.

- *Line 216-217: The authors should highlight before that the R93 mutation has a decreased secretion of FlgD compared to the other single mutations. It might be of interest to determine secretion levels of late substrates as well, e.g. FliC.*

Re: Agreed and mentioned in the text. We do not think it is necessary to determine the secretion levels of late substrates because we do not investigate export specificity of the flagellar protein export apparatus in this study.

- *Line 218-223: The authors mention that chaperone-late substrate complexes bind to FliI, but then they investigate the binding of FlgD to FliI and see no difference. Perhaps it would have been useful to look at binding deficiencies of chaperone-late substrate complexes?*

Re: Thank you so much for your comment. We changed the brief introduction as follows to make it clear:

*“The FliI ATPase plays an important role in substrate recognition²⁷. Addition of purified FliH₂FliI complex at a final concentration of 1.5 μM to the *in vitro* assay solution increases the level of FlgD transported to the inside of inverted membrane vesicles by 20-fold, indicating that the FliH₂FliI complex facilitates the export of FlgD²⁹. We found that the *fliI-5A* mutation abolished the secretion of FlgD in the absence of FliH (Supplementary Fig. 4b), raising the question of whether the *fliI-5A* mutation inhibits the interaction between FliI and FlgD.”*

- *Figure 4B: FliI must have been mislabeled. FliI is indicated as ~37 kDa in size in the blot of the second row (GST-FliHAC), but as 50 kDa in size in all the other blots?*

Re: Corrected

- *Discussion: I found it very difficult to follow all the different mutations in FliH and FliB and their suppressor mutations in various backgrounds and their effect on secretion from previously published papers. Perhaps, these descriptions can be simplified/summarized? I would also consider to place the observations on FliH topology in the results and not solely in the discussion part.*

Re: We simplified our descriptions as much as possible and moved the observations of FliH topology to the result section.

- *The proposed mechanism of the flagellar T3SS is quite speculative and I am not convinced the model is very well supported by the presented data. Certainly, additional data on the dynamic interactions between FliH/FliA/FliH/FliI are needed to substantiate the model in future studies.*

For the present manuscript, my recommendation would be to discuss the open mechanistic questions in more detail. Just to name a few... what are potential functions of the ATPase complex after activation of the PMF-driven protein export? Does the escort function of FliH₂-FliI really play a functional role (e.g. PMID 25068520)? In all the cryo electron tomography images, the ATPase complex appears to be rather stably

associated to the base of the export gate. It might also worthwhile to note that protein export can actually occur in the absence of the ATPase complex (e.g. PMID 25393010, 31911488). What about substrate unfolding via the ATPase as frequently cited as a potential role (e.g. PMID 32346005, 16208377)?

Re: Thank you very much for your comments and suggestions. We rewrote the Discussion based on these comments as much as possible.

Minor comments

- *Line 26: ...mutations conferred a loss-of-function...*

Re: Corrected

- *Line 40-44: References missing*

Re: Added

- *Line 51-52: References missing (e.g. by Kuhlen et al. 2018)*

Re: Here, we decided to cite two review articles as references.

- *Line 61: Missing reference indicated as 14 (Fabiani et al.)*

Re: Cited

- *Line 54-55: Transmembrane domain is misleading, as the proton coupling is thought to be mediated by a cytoplasmic loop between two transmembrane helices and the large cytoplasmic domain of FlhA*

Re: We changed the sentence as follows:

“FlhA assembles into a homo-nonamer through its C-terminal cytoplasmic domain (FlhAc)^{4,5} and forms a pathway for the transit of protons across the cytoplasmic membrane^{6,7}.”

- *Line 66: basal body; C-ring rather than C ring*

Re: Corrected

- *Line 65-68: Include other relevant references*

Re: Minamino and Macnab (2000) have proposed that FlhAc and FlhBc form a docking platform for the cytoplasmic ATPase complex and export chaperones. Therefore, we decided to cite this original paper as a reference.

- *Line 83-86: Sentence is difficult to read.*

Re: We changed this sentence as follows:

“Because FliI-YFP shows turnover between the basal body and the cytoplasmic pool, the FliH₂FliI complex is thought to act as a dynamic carrier to escort FliJ, export substrates and chaperon/substrate complexes from the cytoplasm to the transmembrane export gate complex³⁰.”

- *Line 109: Why did you exclude R76?*

Re: First, my former graduate student could not generate the R76 mutant. Because we constructed FliI-4A and found it is still fully functional when expressed from a pET19-based plasmid, we did not think that it is necessary to construct this R76 single mutant.

- *Supplemental Fig. 1 and Figure 2A, Figure 3 (+ other Figures with motility plates): Did the plates contain supplements to induce the plasmids? According to the Methods section they don't, but I would suggest to include this information in the Figure legend.*

Re: We added only ampicillin to a final concentration of 100 µg/ml. We mentioned it in the figure legends.

- *Supplemental Fig. 2: Indicate that I and H correspond to FliI and FliH*

Re: Mentioned.

- *Figure 4A: What is GST-FliI*?*

Re: GST-FliI* is a degraded product of GST-FliI. We now mentioned it in the legend.

- *Line 231: mutation...FliI*

Re: Corrected

- *Figure 5: Include genotype to the corresponding strain number (e.g. MMH1001 (Δ fliH-fliI)). It would be easier to follow.*

Re: Agreed.

- *Line 281: Reference to Figures doesn't match. Figure 2 and 3 instead?*

Re: Corrected

- *Line 305: FliB*

Re: Corrected

- *Line 309: ...to hydrolyze..*

Re: Corrected

- *Line 343: have shown*

Re: We deleted the first paragraph of the original Discussion section because we had to shorten Discussion as suggested by Reviewer 3.

- *Line 354: ...hetero-hexamer of the...*

Re: Corrected

- *Line 363-364: were over-expressed*

Re: These sentences were deleted to shorten Discussion.

- *Line 395: Figure 5D?*

Re: Corrected

- *Line 396: FliHEN was not specified before.*

Re: Defined FliH_{EN} as the extreme N-terminal region of FliH in text.

- *Line 397: interacts*

Re: Corrected

- *Line 428-432: Consider rephrasing to make it more easy to understand*

Re: We changed the sentence as follows:

“These suggest that conformational rearrangements of FliH_{NCT}, FliH_{Loop} and FliH_{CN} occur in a FliI-dependent manner, allowing the polypeptide channel to be opened and that FliH_{CCT} may regulate the FliH_{NCT}–FliH_{CN} interaction to interfere with premature gate opening.”

- *Line 461: Based on*

Re: Corrected

- *Line 468: ...thereby opening the polypeptide channel...*

Re: Corrected

- *Line 470: through*

Re: Corrected

- *Line 510: inoculated into*

Re: Corrected

To Reviewer #3:

The work by Kinoshita et al aims to analyze the role of positively charged residues at the interface of the flagella T3SS ATPase FliI and FliH in regulating T3S-dependent export.

The paper is well written and the figures are clear. Only the discussion is a bit lengthy (it is a review in itself, not very focussed) and the narrative jumps somewhat between different mutants.

Re: Thank you very much for your supportive comments. We shortened Discussion as much as possible.

General remarks and major points:

The authors show that the simultaneous mutation of five arginines (FliA-5A) impairs secretion through the flagellar T3SS and consequently flagellar motility while the mutation of fewer arginines does not make a difference. The secretion and motility defect of the FliI-5A mutant can be rescued by overexpression of the mutated protein.

FliI is thought to exist in a flagellum-bound hexameric form that shows high ATPase activity and in a flagellum-unbound FliI1-FliH2 complex that shows low ATPase activity. Therefore it was suggested that the binding of FliH to FliI prevents FliI ring formation and thus indirectly ATPase activity. The authors show that the FliI-5A mutation impairs the interaction of FliI with FliH and conclude that the impaired interaction causes the secretion and motility defect. Interestingly, also FliI-2A, FliI-3A and FliI-4A mutations that don't show a defect in motility show the same defect in interaction of FliI and FliH. Consequently, the suggested link from a defect in FliI-FliH interaction to a secretion defect is not conclusive.

Re: As described in our response to Reviewer #1, an interaction between FliI_{EN} and FliH_C is critical for the formation of the FliH₂FliI complex in solution (**Minamino and Macnab. *Mol. Microbiol.* 37, 1494–1503. 2000**), and FliI_{EN} forms not only hydrophobic but also electrostatic interaction networks with FliH_C (**Imada et al. *PNAS* 113, 3633–3638. 2000**). Over-expression of FliI and its mutant variants produced aggregates in the cytoplasm (**new Fig. 3b**), but co-expression with FliH increased their solubility in the cytoplasm (**new Supplementary Fig. 3c**), suggesting that FliH binds to FliI_{EN} of these mutant FliI proteins to interfere with aggregations in the cytoplasm. Because strong ion strength of solvent weakens the electrostatic interactions, we assume that FliH easily dissociates from FliI-2A, FliI-3A, FliI-4A and FliI-5A during Ni affinity chromatography because the binding buffer contains 500 mM NaCl. FliI-2A, FliI-3A and FliI-4A fully restored the motility of the $\Delta fliI$ mutant when they were expressed from pET19b-based plasmids whereas FliI-5A did not. Since over-expression of FliI-5A restored motility to the wild-type level, this suggests that the binding affinity of FliI-5A for FliH_C is weaker than those of FliI-2A, FliI-3A and FliI-4A under physiological conditions.

Furthermore, FliI-5A overexpression cannot compensate for a fliH deletion as the FliI wild type and other mutants can, showing that FliI-5A also has a defect independent

of the FliI-FliH interaction.

Re: Except for the R33A mutant, the other mutants could not compensate for the FliH defect to the wild-type level and especially the R93A mutation reduced the secretion level of FlgD in the absence of FliH (**new Supplementary Fig. 4b**). Because they are fully functional in the presence of FliH, this suggests that these mutant FliI proteins require FliH to fully exert its protein export function.

The authors test whether the FliI-5A defect results from an impaired binding to T3SS substrates, but this is not the case. They then sought to assess whether the FliI-5A mutant showed a reduced binding to the flagellar components FlhA, FlhB and FliJ. The authors claim that no difference in FliI-binding to these proteins is observable between the wild type and the mutant. However, there is no relevant binding observed between any of these proteins at all (maybe with the exception of an almost insignificant FliI-FlhA interaction), precluding the drawing of any conclusions.

Re: The *fliI-5A* mutation did not affect the interactions of FliI with FlgD, FliJ, FlhAc and FlhBc significantly. However, the *fliI-5A* mutation caused a loss-of-function phenotype in the absence of FliH, suggesting that this mutation affects the protein export process after docking of FliI-5A, FliJ and export substrates to the FlhAc-FlhBc docking platform.

The authors then isolated pseudorevertants of the secretion and motility-deficient Δ fliH, FliI-5A mutant that located to the very C-terminus of FlhB. These FlhB mutants substantially rescued the secretion of a Δ fliH, Δ fliI double mutant, suggesting to me that they increase the secretion capacity of the flagellar T3SS in a more general way.

Re: Agreed

Based on all these results the authors hypothesize that an interaction between FliI and FlhB triggers the opening of export gate formed by FliPQR and FlhB. Unfortunately, I can't see the conclusiveness of this hypothesis as the FlhB mutations act independently of FliI altogether. Furthermore, based on our current structural understanding, we have no indication whatsoever that FliI and FlhB might interact. They are likely to reside on different sides of the FlhAc nonamer, unable to interact. No one has shown up to this point that FliI and FlhB interact in an intact system in vivo. I am afraid that weak GST-pulldown data do not suffice to make this point.

Re: FliJ is inserted into the central cavity of the FliI hexamer so that FliJ is brought to the vicinity of the FlhAc ring structure to bind to the linker region of FlhA (FlhAL) connecting FlhAc with the N-terminal transmembrane domain of FlhA, thereby allowing the export gate complex to become an active protein transporter. However, *in situ* structures of the flagellar protein export apparatus have shown that the distance between the FliI and FlhAc ring structures is too long for FliJ to reach FlhAL. Furthermore, the FlhAc ring structure is also too far from the entrance gate. These observations suggest that the *in situ* structures of the export apparatus visualized to date presumably represent the export-off state. Furthermore, the exact location of FlhBc remains unknown although we provided evidence suggesting that FlhBc is actually located in the cytoplasm. It has been reported that the K_D value for the FlhBc-FliI interaction is 84 nM and that ATP hydrolysis by FliI weakens such a strong FlhBc-FliI interaction (**McMurry et al. PLOS One 2015**). Because *in vitro* protein-protein

interactions must reflect certain steps in the flagellar protein export process and the protein export apparatus seems to undergo large conformational changes during the export process, physical communications between FliI and FlhB_C should occur when the export apparatus becomes an active protein transporter. Furthermore, assuming that FlhB_C is located inside the FlhA_C ring structure, physical interactions between FliI and FlhB_C could be also possible because FliI also exists as the FliH₂FliI complex that is thought to act as a dynamic carrier to deliver export substrates to the FlhA_C-FlhB_C docking platform.

We found that over-expression of wild-type FliI activated flagellar protein export by the $\Delta fliH-fliI flhB_{SP2}$ mutant, thereby enhancing the motility to a considerable degree whereas that of FliI-5A did not (**Fig. 5c and d**). Because the *flhB_{SP2}* mutation considerably increased the probability of export substrate entry into the polypeptide channel of the FliP₅FliQ₄FliR₁ complex in the absence of FliH and FliI, this suggests that FliI increases the probability of substrate entry into the polypeptide channel presumably through the interactions between FliI and FlhB_C. Because the *fliI-5A* mutation did not affect the interaction of FliI with FlgD, FliJ, FlhA_C and FlhB_C, we assume that the *fliI-5A* mutation probably inhibits this substrate entry step.

The authors show further that overexpression of the mutants FliI(R33A) and FliI-3A imposes a growth defect on the bacteria that can be rescued by co-overexpression of FliH. Unfortunately, the assaying of the growth defect was done by a barely quantitative plating method and should be repeated with more quantitative assays.

Re: We measured the cell growth of *Salmonella* cells over-expressing FliI(R33A) and FliI-3A and showed that their over-expression caused the growth arrest (**new Fig. 3a**).

The authors postulate that the putative growth defect results from futile ATP hydrolysis caused by an increased ring-forming propensity of these FliI mutants, which is suppressed by FliH-binding. Indeed, an increased ATP hydrolysis was measured in these mutants, however, the data are not fully convincing as very different ATP concentrations were tested to assess the activity of the wild type and the mutant. To improve this dataset, the assays should be repeated at the same ATP concentrations to enable full comparison.

Re: We improved the dataset as suggested by this reviewer. At a FliI concentration of 85 nM, the ATPase activity of wild-type FliI, FliI(R33A) and FliI-3A was 0.008, 1.211, and 1.898 nmol of phosphate min⁻¹ μg⁻¹, respectively, indicating that both FliI(R33A) and FliI-3A exhibits much higher ATPase activities even at a very low protein concentration compared to the wild-type (**new Fig. 3c and Table 1**).

In this section, the authors speculate much on the effect of the tested mutations on FliI ring formation but it is never directly tested. It would be beneficial for the paper to show that FliI ring formation is truly effected, i.e., by size exclusion chromatography or crosslinking.

Re: We carried out *in vitro* reconstruction of the FliI ring structure and observed them by negative stain electron microscopy (**new Fig. 4**). We found that FliI-3A formed a homo-hexamer more efficiently compared to wild-type FliI.

Overall, the presented work analyzes the effects of mutations in FliI and FlhB but

unfortunately, the presented results do not add substantial insight to our current understanding of the T3SS function in general and export gate opening in particular. Thus, I feel that the major conclusions and claims of this paper are overstated. In particular, there is no evidence presented that FliI and FliH interact to open the export gate, as is claimed by the authors.

Re: We agree with this reviewer that there is no direct evidence that the FliI-FliH interaction opens the export gate and therefore toned down our statements as much as possible. However, our genetic and biochemical data suggest that FliI is required for efficient gate opening. So, we would like to keep our hypothesis, and this will be very useful for future studies. Furthermore, we also provided not only direct evidence for a positively charged cluster of FliI being involved in well-regulated FliI ring formation but also an important clue to the energy coupling mechanism of the FliI ATPase. Therefore, we believe this current study adds substantial new insights to our current understanding of the T3SS function.

Minor points:

Page 1 and 2: Title and abstract should be adjusted to reflect the results. Both are much overstated.

Re: We changed the title and abstract as follows:

Title:

“A positive charge cluster of FliI ATPase regulates FliI ring formation for flagellar protein export”

Abstract:

“The FliH₂FliI complex is thought to pilot flagellar subunit proteins from the cytoplasm to the transmembrane export gate complex for flagellar assembly. FliI also forms a homo-hexamer to hydrolyzes ATP, thereby activating the export gate complex to become an active proton-protein antiporter. However, it remains unknown how it occurs. Here we report the role of a positively charged cluster formed by Arg-26, Arg-27, Arg-33, Arg-76 and Arg-93 of FliI in flagellar protein export. Arg-33 and Arg-76 were involved in FliI ring formation. FliI with the R26A/R27A/R33A/R76A/R93A mutations required FliH to fully exert its export function. Gain-of-function mutations in FliH increased the probability of export substrate entry into the export gate complex, thereby restoring the export function of the $\Delta fliH fliI(R26A/R27A/R33A/R76A/R93A)$ mutant. We suggest that the positive charge cluster of FliI is responsible not only for well-regulated hexamer assembly but also for substrate entry into the gate complex.”

Page 4, line 12: “basal” body

Re: Corrected

Page 5, line 10: “chaperone”

Re: Corrected

Page 9, lines 3-10: Please draw a graph of all ATPase assay data.

Re: Agreed and showed the data in new Fig. 3c.

Page 12, line 9: “mutation; “FIII” should be “FII”

Re: Corrected

Page 15 ff: I don’t see the point of this topology assay as the topology is obvious from the solved FlhB structure.

Re: Because the densities corresponding to FlhB_{NCT} and FlhB_C were quite poor in the cryoEM reconstruction, we thought we should carry out the PhoA fusion assay to make sure the subcellular location of FlhB_{NCT} and FlhB_C.

Page 17, line 17: “have” instead of “has”

Re: To shorten Discussion as suggested by this reviewer, we deleted the first paragraph.

Page 43: The model drawn is not correct with respect to the location of the FliPQR FlhB complex.

Re: The locations of FliPQR and FlhB are correct but this cartoon is too simplified reflect them. This schematic diagram is to only explain the gating mechanism of the FliPQR polypeptide channel.

Reviewers' comments:

Reviewer #1 (Remarks to the Author):

Kinoshita et al. substantially revised their manuscript and appropriately addressed my concerns. I recommend publication of the manuscript.

Reviewer #3 (Remarks to the Author):

The authors have addressed several of the reviewer's concerns and suggestions satisfyingly. The data shown in Fig. 3 and Fig. 4 are good examples of this.

Unfortunately, the authors were not able to address the reviewer's concerns on the interpretation of the FliI mutants effects and on the FliI interaction data. In particular, the following comments were not addressed satisfyingly:

Original comment: The authors show that the FliI-5A mutation impairs the interaction of FliI with FliH and conclude that the impaired interaction causes the secretion and motility defect. Interestingly, also FliI-2A, FliI-3A and FliI-4A mutations that don't show a defect in motility show the same defect in interaction of FliI and FliH. Consequently, the suggested link from a defect in FliI-FliH interaction to a secretion defect is not conclusive.

Comment: The response to this concern did not solve the question as it was just based on methodological assumptions but not on hard data.

Original comment: Furthermore, FliI-5A overexpression cannot compensate for a fliH deletion as the FliI wild type and other mutants can, showing that FliI-5A also has a defect independent of the FliI-FliH interaction.

Comment: Unfortunately, the author's response does not address the reviewer's question.

Original comment: The authors test whether the FliI-5A defect results from an impaired binding to T3SS substrates, but this is not the case. They then sought to assess whether the FliI-5A mutant showed a reduced binding to the flagellar components FlhA, FlhB and FliJ. The authors claim that no difference in FliI-binding to these proteins is observable between the wild type and the mutant. However, there is no relevant binding observed between any of these proteins at all (maybe with the exception of an almost insignificant FliI-FlhA interaction), precluding the drawing of any conclusions.

Comment: The problem is not addressed well by the authors. Still, the interaction data do not support the statements made. There is barely any interaction observed between FliI and FlhA, FlhB, or FliJ, making it very difficult to draw any conclusions. Also, much of the discussion is based on the observation of these interactions, for which I do not see much evidence.

In summary, the authors show interesting data that may help to improve our understanding of the flagellar T3SS. However, the overstatement of a number of findings that are not completely conclusive bears the danger of misguiding the field and so these statements should be weakened and openly discussed.

Our responses are listed below.

To Reviewer #1

Kinoshita et al. substantially revised their manuscript and appropriately addressed my concerns. I recommend publication of the manuscript.

Re: Thank you so much for your support for the publication our manuscript in Communications Biology.

To Reviewer #3

The authors have addressed several of the reviewer's concerns and suggestions satisfyingly. The data shown in Fig. 3 and Fig. 4 are good examples of this.

Unfortunately, the authors were not able to address the reviewer's concerns on the interpretation of the FliI mutants effects and on the FliI interaction data. In particular, the following comments were not addressed satisfyingly:

Original comment: The authors show that the FliI-5A mutation impairs the interaction of FliI with FliH and conclude that the impaired interaction causes the secretion and motility defect. Interestingly, also FliI-2A, FliI-3A and FliI-4A mutations that don't show a defect in motility show the same defect in interaction of FliI and FliH. Consequently, the suggested link from a defect in FliI-FliH interaction to a secretion defect is not conclusive.

Original response: As described in our response to Reviewer #1, an interaction between FliI^{EN} and FliH_C is critical for the formation of the FliH₂FliI complex in solution (Minamino and Macnab. *Mol. Microbiol.* **37**, 1494–1503. 2000), and FliI^{EN} forms not only hydrophobic but also electrostatic interaction networks with FliH_C (Imada et al. *PNAS* **113**, 3633–3638. 2000). Over-expression of FliI and its mutant variants produced aggregates in the cytoplasm (new Fig. 3b), but co-expression with FliH increased their solubility in the cytoplasm (new Supplementary Fig. 3c), suggesting that FliH binds to FliI^{EN} of these mutant FliI proteins to interfere with aggregations in the cytoplasm. Because strong ion strength of solvent weakens the electrostatic interactions, we assume that FliH easily dissociates from FliI-2A, FliI-3A, FliI-4A and FliI-5A during Ni affinity chromatography because the binding buffer contains 500 mM NaCl. FliI-2A, FliI-3A and FliI-4A fully restored the motility of the $\Delta fliI$ mutant when they were expressed from pET19b-based plasmids whereas FliI-5A did not. Since over-expression of FliI-5A restored motility to the wild-type level, this suggests that the binding affinity of FliI-5A for FliH_C is weaker than those of FliI-2A, FliI-3A and FliI-4A under physiological conditions.

Comment: The response to this concern did not solve the question as it was just based on methodological assumptions but not on hard data.

Re: We agree with this reviewer regarding this point. Because the *fliI-2A*, *fliI-3A* and *fliI-4A* mutations significantly reduced the binding affinity of FliI for FliH, but did not affect flagella-driven motility at all, the interaction between FliH and the positive charge cluster of FliI is dispensable for flagellar protein export. However, because a complete loss of these positive charges by *fliI-5A* mutation caused a reduction in

motility (**See Fig. 3 in the revised MS**), the positive charge cluster of FliI is presumably required for the interaction with other export apparatus components for efficient flagellar protein export. Therefore, we changed our description as follows:

“Therefore, we conclude that electrostatic interactions between the positive charge cluster of FliI_N and FliH_C are dispensable for flagellar protein export and assembly although these interactions significantly stabilize an interaction between FliI_{EN} and FliH_C. Because a complete loss of the positive charges by *fliI-5A* mutation reduced the flagellar type III protein transport activity (Fig. 3**), we hypothesize that this positive charge cluster may be involved in the interaction with other export apparatus components for efficient flagellar protein export process.”**

Original comment: Furthermore, FliI-5A overexpression cannot compensate for a *fliH* deletion as the FliI wild type and other mutants can, showing that FliI-5A also has a defect independent of the FliI-FliH interaction.

Original response: Re: Except for the R33A mutant, the other mutants could not compensate for the FliH defect to the wild-type level and especially the R93A mutation reduced the secretion level of FlgD in the absence of FliH (**new Supplementary Fig. 4b**). Because they are fully functional in the presence of FliH, this suggests that these mutant FliI proteins require FliH to fully exert its protein export function.

Comment: Unfortunately, the author’s response does not address the reviewer’s question.

Re: We agree with this reviewer that FliI-5A reduces the flagellar protein transport activity independently of the FliH-FliI interaction. As described in our previous response letter, except for FliI(R33A), single alanine substitutions in the positive charge cluster significantly reduced motility in soft agar and the secretion level of FlgD when FliH is missing (**Fig. 6a left panel of the revised manuscript, also attached below**). Furthermore, in the absence of FliH, the R26A/R27A double mutation (FliI-2A) significantly reduced flagella-driven motility compared to the R26A and R27A single mutations, and the additional R33A mutation (FliI-3A) improved the motility defect of the R26A/R27A mutant to a considerable degree (**Fig. 6a right panel**). Interestingly, a complete loss of the positive charges of FliI by *fliI-5A* mutation inhibited the secretion of FlgD (**Fig. 6b**). This suggests that Arg-26, Arg-27, Arg-33 and Arg-93 in the positive charge cluster of FliI are involved in well-regulated flagellar protein export.

Fig. 6a, b of the revised manuscript

Original comment: The authors test whether the FliI-5A defect results from an impaired binding to T3SS substrates, but this is not the case. They then sought to assess whether the FliI-5A mutant showed a reduced binding to the flagellar components FlhA, FlhB and FliJ. The authors claim that no difference in FliI-binding to these proteins is observable between the wild type and the mutant. However, there is no relevant binding observed between any of these proteins at all (maybe with the exception of an almost insignificant FliI-FlhA interaction), precluding the drawing of any conclusions.

Original response: The *fliI-5A* mutation did not affect the interactions of FliI with FlgD, FliJ, FlhAc and FlhBc significantly. However, the *fliI-5A* mutation caused a loss-of-function phenotype in the absence of FliH, suggesting that this mutation affects the protein export process after docking of FliI-5A, FliJ and export substrates to the FlhAc-FlhBc docking platform.

Comment: The problem is not addressed well by the authors. Still, the interaction data do not support the statements made. There is barely any interaction observed between FliI and FlhA, FlhB, or FliJ, making it very difficult to draw any conclusions. Also, much of the discussion is based on the observation of these interactions, for which I do not see much evidence.

Re: The K_D value for the FlhBc-FliI interaction has been estimated to be 84 nM (McMurry *et al.* *PLOS One* 2015), indicating a strong interaction between FlhBc and FliI. In contrast, our pull-down assays by GST affinity chromatography showed only a weak interaction of FliI with FlhBc presumably because of a very fast dissociation rate. When we carefully compared the amounts of FliI and FliI-5A co-purified with GST-FlhBc to the GST control, FliI and FliI-5A showed some interactions with GST-FlhBc even though the amounts of FliI and FliI-5A co-purified with GST-FlhBc were rather small (Fig. 6d of the revised manuscript, also attached below). Furthermore, compared to the GST control, when loaded with GST-FlhBc, elution of FliI and FliI-5A

was clearly observed as a delayed wash out (**Fig. 6d**), reflecting weak and highly dynamic interactions of FliI with GST-FlhB_C (Note: Pull-down assay is a nonequilibrium method, and therefore if the off rate is high enough, extensive washing may not only result in a significant loss of the binding partner but also in a significantly delayed elution of the binding partner during the extensive washing process). Highly dynamic interactions were also observed in the FliI-FlhA_C and FliI-FliJ interactions as judged by our pull-down assays (**Fig. 6d**). Therefore, we think that relevant bindings are actually observed between these proteins. We carefully compared a set of data obtained by pull-down assays and found that the amount of FliI-5A co-eluted with GST-FlhB_C was significantly higher than that of wild-type FliI and that there were no apparent differences in the binding affinities between FliI and FliI-5A for FlhA_C and FliJ (**Fig. 6d**).

Fig. 6d of the revised manuscript

Interestingly, the level of FlgD secreted by the $\Delta fliH fliI-5A$ mutant was lower than that by the $\Delta fliH-fliI$ double null mutant (**Fig. 6b of the revised manuscript**, also attached above). Furthermore, over-expression of FliI-5A reduced the secretion level of FlgD by the $\Delta fliH-fliI flhB_{SP2}$ mutant compared to the vector control (**Fig. 7d of the revised manuscript**, also attached below). These observations indicate that FliI-5A exerts an inhibitory effect on flagellar protein export in the absence of FliH. Since the binding affinity of FliI-5A for FlhB_C was higher than that of wild-type FliI, and FliI-5A retained the ability to bind to FlgD, we suggest that the docking of FliI-5A to FlhB_C inhibits the flagellar protein export process in the absence of FliH. Therefore, we propose that the positive charge cluster of FliI may regulate the binding affinity of FliI for FlhB_C to facilitate the subsequent entry of flagellar building blocks into the polypeptide channel of the export gate complex and that the dissociation of FliI from FlhB_C may be required for efficient substrate entry into the polypeptide channel. Because FliI-5A is functional in the presence of FliH but not in its absence, we also propose that the interaction between FliH_C and FliI_{EN} is required for the positive charge cluster of FliI to undergo its proper conformational changes coupled with the substrate entry process to modulate the binding affinity of FliI for FlhB_C.

Fig. 7d of the revised manuscript

In summary, the authors show interesting data that may help to improve our understanding of the flagellar T3SS. However, the overstatement of a number of findings that are not completely conclusive bears the danger of misleading the field and so these statements should be weakened and openly discussed.

Re: We agree with this reviewer and softened our claim to avoid misleading the T3SS field as much as possible. To do so, we completely deleted the last paragraph describing our possible model from the text and our model figure (Fig. 7 of the previous version of our manuscript).

Reviewers' comments:

Reviewer #1 (Remarks to the Author):

The role of the ATPase FliI in protein export via T3SS remains elusive and the present manuscript by Kinoshita and colleagues provides important new insights into the molecular function of the T3SS of the bacterial flagellum.

In the present, second revision of their manuscript, the authors addressed the remaining concerns by addition of supporting data to the main manuscript, which greatly improved the presentation of their results. They further toned-down their conclusions concerning the role of the positive charge cluster in FliI and possible implication on the molecular mechanism of protein export via the T3SS. Their responses to the raised concerns in their rebuttal letter are also convincing.

Accordingly, Kinoshita et al. appropriately addressed the remaining concerns in the present revision and I strongly support publication of the manuscript in *Communications Biology*.

Reviewer #4 (Remarks to the Author):

Kinoshita and colleagues investigated the importance of a positively charged cluster of 5 arginine residues (R26/R27/R33/R76/R93) present in FliI. The authors found that mutation to alanine of all of these residues leads to defects in ring formation and flagellar protein export. In addition, the authors report that mutant forms of FliI with 2-A;3-A;4-A or 5-A replacements have lower affinity for FliH. Furthermore, the authors found that strong overexpression of the FliI-R33A and the FliI-3A (R26A/R27A/R33A) mutant forms significantly reduced cell growth, which was rescued by co-deleting FliH. They speculate that this result relates with an increase in the stabilization of FliI-R33A and FliI-R26A/R27A/R33A levels in the absence of FliH, which was also accompanied by an increase in ring formation. Despite having observed different ATPase activities for their FliI mutants, the authors report that the mutation of all 5 arginines increased the binding of FliI to FlhB(C) without altering its affinity for FlgD, FlhA(C) and FliJ. Based on these results the authors propose that binding of FliI-5A to FlhB(C) favors the inhibition of flagellar protein export in the Δ fliH fliI-5A mutant. Next, the authors screened for pseudorevertants from the Δ fliH fliI-5A mutant, finding that two insertion mutations of flhB can at least partially rescue motility of the parental line, but not of flagellar protein export. Finally, using PhoA fusion assays the authors conclude that the FlhB(NCT), FlhB(C) and FlhA(NCT) regions are located in the cytoplasm, which leads them to suggest that these protein regions are near to each other in the cytoplasm when functioning at the export gate complex.

While the concepts proposed in the manuscript are interesting, the authors over interpret results, making claims that are not fully supported by their data or for which the data is not appropriately quantified.

Major comments:

1 - In contrast to the clearly detectable FliI-FliH association analyzed in Sup.Fig2., the interactions analyzed in Fig. 6D seem too weak to support such strong mechanistic conclusions made by the authors. Furthermore, the contrast/brightness of the negative controls for pull downs using a GST-binding column in the top panel seem off when compared to the actual experiments that follow below. This should be corrected as there seems to be some FliI protein signal in the E1 fraction on the top panels, meaning that it binds nonspecifically to GST or the column under the experimental conditions used. Normally a small amount wouldn't be a problem; however, as the interactions analyzed below are considerably weak this makes this biochemical data highly unreliable. I suggest that triplicates of the blots in Fig. 6D are quantified so that the potential differences claimed by the authors can be better determined based on statistical analysis.

2 - Furthermore, with the exception of the OD and the ATPase activity, no quantifications are done for the remaining data in the manuscript. I strongly suggest that the authors quantify all of the data on the manuscript (normalizing to controls whenever possible) so that their claims can be better supported by actual numerical analyses of the data.

Minor comments:

3 - In Fig. 4B, the top and the bottom lanes analyze FliI in the same samples, only using different detection methods (staining vs a specific antibody). However, to this reviewer, even when one takes in consideration that the two methods have different sensitivities, the trend of the samples is clearly not the same in the top vs bottom lanes. Are these examples the most representative the authors have?

4 - The number of repeats that were carried out for each gel and biochemical experiment shown in the manuscript should be clearly stated.

5 - No statistical analysis is done for any of the datasets in the manuscript.

6 - Line 22 – typo: “to hydrolyzes ATP”;

Our responses are listed below. We highlighted all changes in red in the revised manuscript.

To Reviewer #1

The role of the ATPase FliI in protein export via T3SS remains elusive and the present manuscript by Kinoshita and colleagues provides important new insights into the molecular function of the T3SS of the bacterial flagellum.

In the present, second revision of their manuscript, the authors addressed the remaining concerns by addition of supporting data to the main manuscript, which greatly improved the presentation of their results. They further toned-down their conclusions concerning the role of the positive charge cluster in FliI and possible implication on the molecular mechanism of protein export via the T3SS. Their responses to the raised concerns in their rebuttal letter are also convincing.

Accordingly, Kinoshita et al. appropriately addressed the remaining concerns in the present revision and I strongly support publication of the manuscript in Communications Biology.

Re: Thank you so much for your strong support for publication of our manuscript in Communications Biology.

To Reviewer #4

Kinoshita and colleagues investigated the importance of a positively charged cluster of 5 arginine residues (R26/R27/R33/R76/R93) present in FliI. The authors found that mutation to alanine of all of these residues leads to defects in ring formation and flagellar protein export. In addition, the authors report that mutant forms of FliI with 2-A;3-A;4-A or 5-A replacements have lower affinity for FliH. Furthermore, the authors found that strong overexpression of the FliI-R33A and the FliI-3A (R26A/R27A/R33A) mutant forms significantly reduced cell growth, which was rescued by co-deleting FliH. They speculate that this result relates with an increase in the stabilization of FliI-R33A and FliI-R26A/R27A/R33A levels in the absence of FliH, which was also accompanied by an increase in ring formation. Despite having observed different ATPase activities for their FliI mutants, the authors report that the mutation of all 5 arginines increased the binding of FliI to FlhB(C) without altering its affinity for FlgD, FlhA(C) and FliJ. Based on these results the authors propose that binding of FliI-5A to FlhB(C) favors the inhibition of flagellar protein export in the Δ fliH fliI-5A mutant. Next, the authors screened for pseudorevertants from the Δ fliH fliI-5A mutant, finding that two insertion mutations of flhB can at least partially rescue motility of the parental line, but not of flagellar protein export. Finally, using PhoA fusion assays the authors conclude that the FlhB(NCT), FlhB(C) and FlhA(NCT) regions are located in the cytoplasm, which leads them to suggest that these protein regions are near to each other in the cytoplasm when functioning at the export gate complex.

While the concepts proposed in the manuscript are interesting, the authors over interpret results, making claims that are not fully supported by their data or for which the data is not appropriately quantified.

Re: Thank you so much for your comments. We quantified our datasets as much as possible.

Major comments:

1 - In contrast to the clearly detectable FliI-FliH association analyzed in Sup.Fig2., the interactions analyzed in Fig. 6D seem too weak to support such strong mechanistic conclusions made by the authors. Furthermore, the contrast/brightness of the negative controls for pull downs using a GST-binding column in the top panel seem off when compared to the actual experiments that follow below. This should be corrected as there seems to be some FliI protein signal in the E1 fraction on the top panels, meaning that it binds nonspecifically to GST or the column under the experimental conditions used. Normally a small amount wouldn't be a problem; however, as the interactions analyzed below are considerably weak this makes this biochemical data highly unreliable. I suggest that triplicates of the blots in Fig. 6D are quantified so that the potential differences claimed by the authors can be better determined based on statistical analysis.

Re: We carried out three independent measurements and quantified triplicates of the immunoblots as suggested (**Fig 7 of the revised manuscript, also attached below**). The *fliI-5A* mutation significantly increased the binding affinities of FliI for FliH_C and FliJ. In contrast, this *fliI-5A* mutation slightly reduced the binding affinity for FliH_A because the elution profile of FliI-5A with GST-FliH_A was almost the same as the GST control. Based on these new observations, we modified our text as follows:

Fig. 7 of the revised manuscript

“The amounts of Flii-5A co-purified with GST-FliH_{B_C} and GST-FliJ were higher than those of wild-type Flii (Fig. 7b), indicating that the *flii-5A* mutation increases the binding affinities of Flii for FliH_{B_C} and FliJ. In contrast, the *flii-5A* mutation reduced the binding affinity of Flii for FliH_{A_C} (Fig. 7b). Because the level of FlgD secreted by the $\Delta fliH flii-5A$ mutant was lower than that by the $\Delta fliH-flii$ double null mutant (Fig. 3d), we suggest that Flii-5A may bind to FliH_{B_C} and FliJ to block the flagellar protein export process in the absence of FliH. Since the over-expression of Flii-5A restored motility to the wild-type level in the presence of FliH (Fig. 3a), we propose that an interaction between the positive charge cluster of Flii and FliH_{A_C} may be involved in flagellar protein export.”

2 - Furthermore, with the exception of the OD and the ATPase activity, no quantifications are done for the remaining data in the manuscript. I strongly suggest that the authors quantify all of the data on the manuscript (normalizing to controls whenever possible) so that their claims can be better supported by actual numerical analyses of the data.

Re: We quantified our datasets as much as possible.

Minor comments:

3 - In Fig. 4B, the top and the bottom lanes analyze FliI in the same samples, only using different detection methods (staining vs a specific antibody). However, to this reviewer, even when one takes in consideration that the two methods have different sensitivities, the trend of the samples is clearly not the same in the top vs bottom lanes. Are these examples the most representative the authors have?

Re: To avoid confusion, we have removed the CBB stained gel from the figure.

4 - The number of repeats that were carried out for each gel and biochemical experiment shown in the manuscript should be clearly stated.

Re: We mentioned this point in each figure legend.

5 - No statistical analysis is done for any of the datasets in the manuscript.

Re: We carried out statistical analyses for our datasets.

6 - Line 22 – typo: “to hydrolyzes ATP”;

Re: Corrected

REVIEWERS' COMMENTS:

Reviewer #4 (Remarks to the Author):

In the current revised version of the manuscript, Kinoshita and colleagues do a reasonable effort at addressing most of my concerns by adding several quantifications of their data and improving their examples. However, I still did not find any information regarding the statistical analysis of their data. The authors often use the word "significantly" to describe differences observed but they do not mention which statistical tests were used and whether such differences are Statistically significant or not... This information should be added to the manuscript methods/figures/legends for it to be suitable for publication in Communications Biology

Our responses are listed below.

To Reviewer #4

In the current revised version of the manuscript, Kinoshita and colleagues do a reasonable effort at addressing most of my concerns by adding several quantifications of their data and improving their examples. However, I still did not find any information regarding the statistical analysis of their data. The authors often use the word "significantly" to describe differences observed but they do not mention which statistical tests were used and whether such differences are Statistically significant or not... This information should be added to the manuscript methods/figures/legends for it to be suitable for publication in Communications Biology

Re: Thank you so much for your support for publication of our manuscript in Communications Biology. We carried out two-tailed Student's *t*-test and provided *p*-values in the Figures and Figure legends and in the results section. We really appreciate this reviewer for all helpful comments and suggestions for improving our manuscript.